# ALPL-1 is a target for chimeric antigen receptor therapy in osteosarcoma

Nadia Mensali [1], Hakan Köksal[1,20], Sandy Joaquina [1,20], Patrik Wernhoff [1], Nicholas P. Casey[1], Paola Romecin[2,3], Carla Panisello [2,3], René Rodriguez [4,5,6], Lene Vimeux[7], Asta Juzeniene[8], Marit R. Myhre[1], Anne Fåne[1], Carolina Castilla Ramírez[9], Solrun Melkorka Maggadottir[1], Adil Doganay Duru [10], Anna-Maria Georgoudaki[10,11], Iwona Grad[8], Andrés Daniel Maturana[12], Gustav Gaudernack[13], Gunnar Kvalheim[1], Angel M. Carcaboso [14], Enrique de Alava [9,15], Emmanuel Donnadieu [7], Øyvind S. Bruland[16], Pablo Menendez[2,3,17,18,19], Else Marit Inderberg[1] ✉ & Sébastien Wälchli [1] ✉

Osteosarcoma (OS) remains a dismal malignancy in children and young adults, with poor outcome for metastatic and recurrent disease. Immunotherapies in OS are not as promising as in some other cancer types due to intra-tumor heterogeneity and considerable off-target expression of the potentially targetable proteins. Here we show that chimeric antigen receptor (CAR) T cells could successfully target an isoform of alkaline phosphatase, ALPL-1, which is highly and specifically expressed in primary and metastatic OS. The target recognition element of the second-generation CAR construct is based on two antibodies, previously shown to react against OS. T cells transduced with these CAR constructs mediate efficient and effective cytotoxicity against ALPL-positive cells in in vitro settings and in state-of-the-art in vivo orthotopic models of primary and metastatic OS, without unexpected toxicities against hematopoietic stem cells or healthy tissues. In summary, CAR-T cells targeting ALPL-1 show efficiency and specificity in treating OS in preclinical models, paving the path for clinical translation.

Osteosarcoma (OS) is an aggressive malignant tumor, mainly affecting children and young adults. Although it is a rare disease in the overall population, OS represents 5–10% of pediatric and adolescent cancer patients. RARECARE has estimated between 0.3 and 0.5 cases of OS being diagnosed per 100.000 individuals per year in the European Union[1]. Approximately 4.4 cases of OS per million children are reported annually[2].The introduction of combinatory chemotherapy greatly improved patient survival, but a survival plateau has been achieved, beyond which further improvements have proven difficult[3,4].

Primary metastatic OS still carries a dismal prognosis and lung metastases continue to be the major cause of death[5,6]. The hematogenous spread of circulating tumor cells is an early event also in the majority of OS patients assumed to have localized disease, with formation of micrometastases in lungs and bone marrow (BM)[7] that will later manifest as chemotherapy-resistant overt metastases. Thus, there is a clear need to identify novel treatments targeting such micrometastases. The clinical translation of strategies proposed to block this cancer cell migration has so far been discouraging[8].

Immunotherapy has remarkable success in some cancer types[9] and different strategies have been tested in OS with so far disappointing clinical benefit[10–12]. More innovative solutions such as chimeric antigen receptor (CAR) T-cell therapy are presently tested clinically[13].

The specificity of a CAR molecule resides in its antigen binding domain, which is typically made of an antibody-derived molecule known as single chain variable fragment (scFv). It is therefore a

prerequisite to identify selective and specific cancer markers expressed at the tumor cell surface to generate an efficient CAR. A few antigens have been targeted in pre-clinical and clinical settings to generate anti-OS CAR molecules, including human epidermal growth factor receptor-2 (HER-2), which was shown to be safe and well tolerated in a Phase I sarcoma basket trial[14]. Unfortunately, HER-2 is expressed by only a subset of OS samples. At least three additional targets (GD2, EGFR and B7-H3) are being tested in the clinic but no results are yet available[8,15,16].

Three decades ago two monoclonal antibodies, TP-1 and TP-3, were generated by inoculating mice with a human OS cell line derived from a patient with primary lung metastases[17]. These antibodies are selective for OS with lower reactivity for other sarcoma types[18]. They exhibit limited cross-reactivity to healthy tissues, and can discriminate OS cells in patient BM samples[18]. The antibodies bind unique epitopes of the same cell surface-expressed alkaline phosphatase (ALP) on OS cells. ALP is an enzyme family composed of 4 different members[19] and the detection of serum ALP activity is a marker for different diseases including OS. As TP antibody binding signal did not correlate with the activity of serum ALP detected by other anti-ALP diagnostic antibodies[18], another isozyme could be the target but remained unidentified. Importantly, TP-1 antibody was injected into five OS patients for immunoscintigraphy studies and demonstrated to detect lung metastases[17,20]. This was also true for immune-PET based on TP-3 in dogs with spontaneous OS[21], suggesting that their target epitopes are OS specific and that the clinical use of these antibodies could be safe.

Here, we identify an ALP member, ALPL-1, as the bona fide target of TP-1 and TP-3 antibodies and confirm their specificity for OS with cross-reactivity limited to proliferating osteoblasts. Two CAR molecules, OSCAR-1 and OSCAR-3, designed based upon the coding sequences of the TP hybridomas, efficiently redirect immune cell against OS. We demonstrate pre-clinical proof-of-concept of OSCAR T cells which are effective and not cross-reactive. Considering the distribution of ALPL-1, OSCAR T cells emerge as a concievable cellular therapy solution for a large proportion of OS patients.

## Results

### TP-1 and TP-3 antibodies recognize surface expressed ALPL-1
TP-1 and TP-3 antibodies were isolated from hybridomas created from a mouse injected with a human OS cell line derived from a patient tumor metastasis, TPX[17]. The antibody specificity was suggested to be against a molecule with an alkaline phosphatase activity[18]. To discover the precise identity of the antigen we used the Retrogenix platform screening a collection of nearly 5000 different cDNAs encoding for membrane proteins. The same target was extracted in two successive rounds for the two TP antibodies and confirmed in the last panel (Supplementary Fig. 1a; left). The target was identified as the *ALPL* gene (NM_000478.6) isoform-1 (NP 000469.3). Interestingly, the isoform-3 (NP 001120973.2) which was also included in the screen and only differs by 55 amino acids from isoform-1, was not recognized by TP antibodies. We also noticed that the IGSF-1 protein was weakly recognized by TP-1 but not by TP-3 in this screen. The Human Protein Atlas (https://www.proteinatlas.org/ENSG00000147255-IGSF1) reports that IGSF-1 is mainly present in brain tissue and can be detected at high levels in HepG2 cells, and to a lower extent in HeLa and U2OS cell lines. Although HepG2 cells were confirmed positive for IGSF-1 by Western blotting (Supplementary Fig. 1a; middle), they were negative for TP-1 staining (Supplementary Fig. 1a; right). Since TP antibodies do not bind ALPL in Western blot (Supplementary method 1) this was tested by flow cytometry. We thus concluded that the signal reported for IGSF-1 in the Retrogenix platform screening was non-specific and that TP-1, like TP-3, was ALPL-1 restricted.

We confirmed ALPL-1 specificity by overexpression and knockout experiments. As shown in Fig. 1a, both TP-1 and TP-3 reacted against HEK-293 cells transfected with the *ALPL-1* cDNA, but not against the parental cell line. As a detection control, we used OHS, an OS cell line previously described to be stained by the TP antibodies[22]. We further validated these results by knocking out the *ALPL* gene using CRISPR/CAS9 technology in OHS cells (Fig. 1b, c) and observed that both TP antibodies, as well as a commercial anti-ALPL antibody, did not bind knockout OHS cells. Taken together, these data demonstrate that TP antibodies are specific for unique epitopes residing on the antigen, ALPL-1, confirming previous propositions[18] that the TP target was an alkaline phosphatase of a molecular weight of around 80 kD (Fig. 1c). Different OS cell lines were previously stained with TP antibodies[18], and we repeated this experiment on most of the cell lines used in the present study. We observed that the antibody signal varied in intensity, but all OS cell lines were positive (Fig. 1d). Importantly, although targeted against the same antigen, previous publications reported lower staining intensity of TP-1 compared to TP-3. Since ALPL-1 was herein identified as the bona fide target for the TP antibodies, we tested their binding capacities which were found similar, and we confirmed the higher staining intensity of TP-3 (Fig. 1d).

### ALPL-1 is highly expressed in sarcoma samples
It was previously shown that TP antibodies specifically reacted against virtually all OS samples tested and to some extent against other types of sarcoma tissues[17]. Although the gene *ALPL* codes for a non-tissue specific alkaline phosphatase, the presence of the protein at the cell surface seems to be restricted to a limited number of tissues. We thus undertook an analysis of ALPL-1 expression in next-generation sequencing (NGS) datasets (Supplementary Table 1) for gene expression, which confirmed that sarcomas, in general, expressed *ALPL-1* with an increased level in OS samples (Supplementary Fig. 1b, c). We then studied the presence of *ALPL-1* in relevant healthy tissues (bone, lungs) compared with OS lung metastasis samples, and detected a significant increase in diseased tissues (Fig. 1e). From this data, we confirmed the increased presence of *ALPL-1* mRNA in OS samples supporting the initial observations of the membrane protein ALPL-1 using TP antibodies[17]. The data were corroborated by flow cytometry staining of ALPL-1 protein. Here we used TP-3 antibody since it appeared more sensitive than TP-1 to detect ALPL-1 at the cell surface. We confirmed previous data where strong positivity in all model cell lines except one was detected, U2OS[18] (Fig. 1f, open circle), and an even more dramatic staining in OS patient derived xenograft (OS PDX) cells. Finally, mesenchymal stem cells (MSC) which can be derived in osteoblast were negative. We also studied the antigen density of 3 OS cell lines and non-OS cells and observed a complete negativity in the latter ones (Supplementary Fig. 1d). We also checked expression of ALPL-1 by immunohistochemistry (IHC) on OS and normal tissues (pancreas, stomach, uterus, lung and kidney). We detected a faint staining in some endothelia and the proximal renal tubules, in striking contrast with the OS samples, showing a high cytoplasmic and membrane staining (Fig. 1g). We cannot exclude the fact that some healthy tissues are also ALPL-1 positive, but antibody staining was negative[17], suggesting that the protein might be membrane-bound only in specific tissues. Thus, ALPL-1 seems to be a moving target which is located at the cell membrane only in certain conditions, and this can obviously not be distinguished by RNA analysis.

### TP-1 and TP-3 derived CARs redirect primary T cells
We used previously published hybridoma sequences[23] to design scFv, and subcloned them into a second-generation CAR backbone[24] linked by a 2A skipping peptide to a truncated CD34 protein[25], in order to detect transduced cells (Fig. 2a, top). TP-1 and TP-3 derived CARs were named OSCAR-1 and OSCAR-3, respectively. Primary T cells from healthy donors were transduced with OSCARs and the presence of the constructs was confirmed by staining with anti-murine Fab and anti-human CD34 antibodies (Fig. 2a, bottom). As shown, OSCAR-1 was not properly detected by the anti-Fab, but the strong CD34 signal

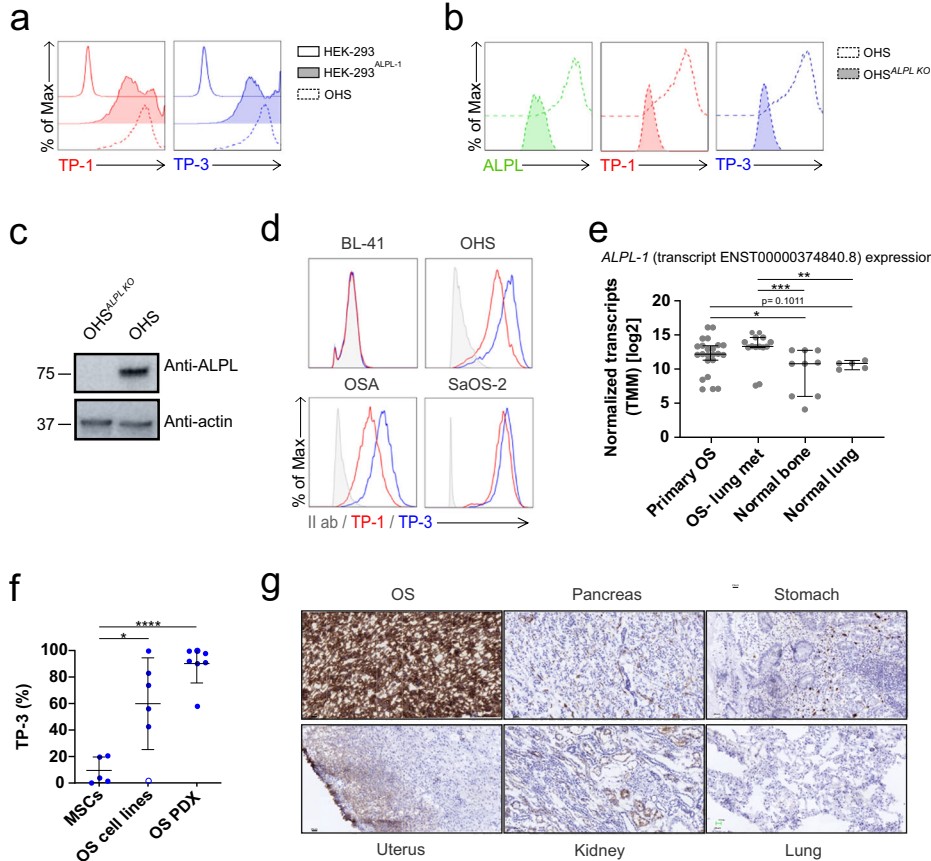

**Fig. 1 | TP antibody characterization. a** Representative TP-1 (left) and TP-3 (right) flow histograms for wild-type HEK-293 (top), HEK-293 transfected with ALPL-1 cDNA (HEK-293^ALPL-1) (middle), and OHS cells (bottom). **b** Representative ALPL (left), TP-1 (middle) and TP-3 (right) flow histograms for ALPL knock out OHS (OHS^ALPL KO) and parental OHS. **c** Western Blotting of OHS^ALPL KO cells shown in **b** to confirm ALPL knock out. Source data are provided as a Source Data file. **d** Representative TP-1 (red) and TP-3 (blue) flow histograms for OS tumor cell lines OHS, OSA and SaOS-2. BL-41 cells (TP negative target). Secondary antibody control (grey). **e** Expression of *ALPL-1* transcript ENST00000374840.8 in primary (*n* = 9) and metastatic (met) OS (*n* = 5), and in normal bone (*n* = 24) and lung tissues (*n* = 14). Counts were normalized by Trimmed mean of M values (TMM) method

prior to log2 transformation. Statistical comparison was performed using two-tailed Mann-Whitney test. (ns, not significant $p > 0.05$, *$p = 0.0376$, **$p < 0.01$, ***$p < 0.001$). Data are presented as median with 95% confidence interval. **f** Flow cytometry ALPL-1 protein expression in MSCs, OS cell lines (143B, G-292, U2OS, OST-3, MG-63 and SAOS-2) and PDXs (*n* = 7) detected with TP-3 antibody. U2OS is open circle. Data are shown as mean ± s.d. Statistical comparisons were performed with two-tailed unpaired Student *t*-test (ns not significant $p > 0.05$, *$p < 0.05$, ****$p < 0.0001$). **g** Immunohistochemistry staining for ALPL-1 using TP-3 antibody on osteosarcoma (OS) and normal tissues (pancreas, stomach, uterus, lung and kidney). Images 20X, 25 micron bar.

indicated that the cells were efficiently transduced, thus, OSCAR-1 is either less stable or less detected. We also monitored the expansion capacity of OSCAR T cells compared to non-transduced T cells (mock) upon CD3/CD28 bead stimulation and we found that OSCAR T cells expanded similarly to mock T cells (Fig. 2b), overall suggesting that the constructs were neither toxic nor cross-reactive against T cells and could be manufactured for clinical use. We first verified that OSCARs conserved TP-antibody specificity against ALPL-1. To this end, we performed bioluminescence (BLI)-based killing assays where OSCAR T cells were co-incubated with HEK-293 cells, transfected or not with ALPL-1 cDNA. The killing activity was only detected in the presence of ALPL-1 (Fig. 2c). We also used *ALPL* knockout OHS cells (Fig. 1b) and observed that the removal of the target had a protective effect against OSCAR-directed killing (Fig. 2c), supporting that ALPL-1 was indeed the target for OSCAR T cells. This was further confirmed when additional OS cell lines staining positively with TP antibodies (Fig. 1d) were shown to be sensitive to OSCAR T cells (Fig. 2d). Altogether, these data confirm OSCARs' specificity for cells expressing ALPL-1 at their surface.

Next, we analyzed whether OSCARs could evoke a sufficiently strong stimulation of the expressing T cells to trigger cytokine release. We measured the level of selected cytokines in the supernatant of

OSCAR T cells after overnight incubation with OS cell lines. As shown, OS cell lines could stimulate OSCAR but not mock T cells (Fig. 2e). Interestingly, OSCAR-3 seems to be slightly more potent in triggering the release of some of the cytokines. IL-2 was produced at lower amounts by OSCAR-1 T cells; however cytotoxicity was similar between the two constructs. In addition, we also observed that the cytokine release was not correlated to the intensity of ALPL-1 at the cell surface, although OHS had 4 times more ALPL-1 density compared to OSA (Supplementary Fig. 1d), the cytokine release triggered by the two cell lines was similar (Fig. 2e). Next, we assessed the ability of OSCAR T cells to withstand repeated antigen encounter, in which cytolytic function and proliferation capacity were monitored upon ALPL-1 antigen stimulation every 7 days for a total of 3 rechallenges. Our results indicate that OSCAR T cells maintained robust short-term effector function as they mediated effective elimination of OS cells until day 21 (Fig. 2f). CAR T cells also maintained their proliferative capacity up to the 2nd rechallenge (Supplementary Fig. 2a). In order to confirm that the OSCAR T cells proliferated in a antigen-dependent manner, we repeated a rechallenge with 4 healthy donors in which the OSCAR or irrelevant CAR (anti-CD19, CD19CAR) T cells were grown on HEK cells expressing or not ALPL-1. We monitored T cell proliferation by CTV

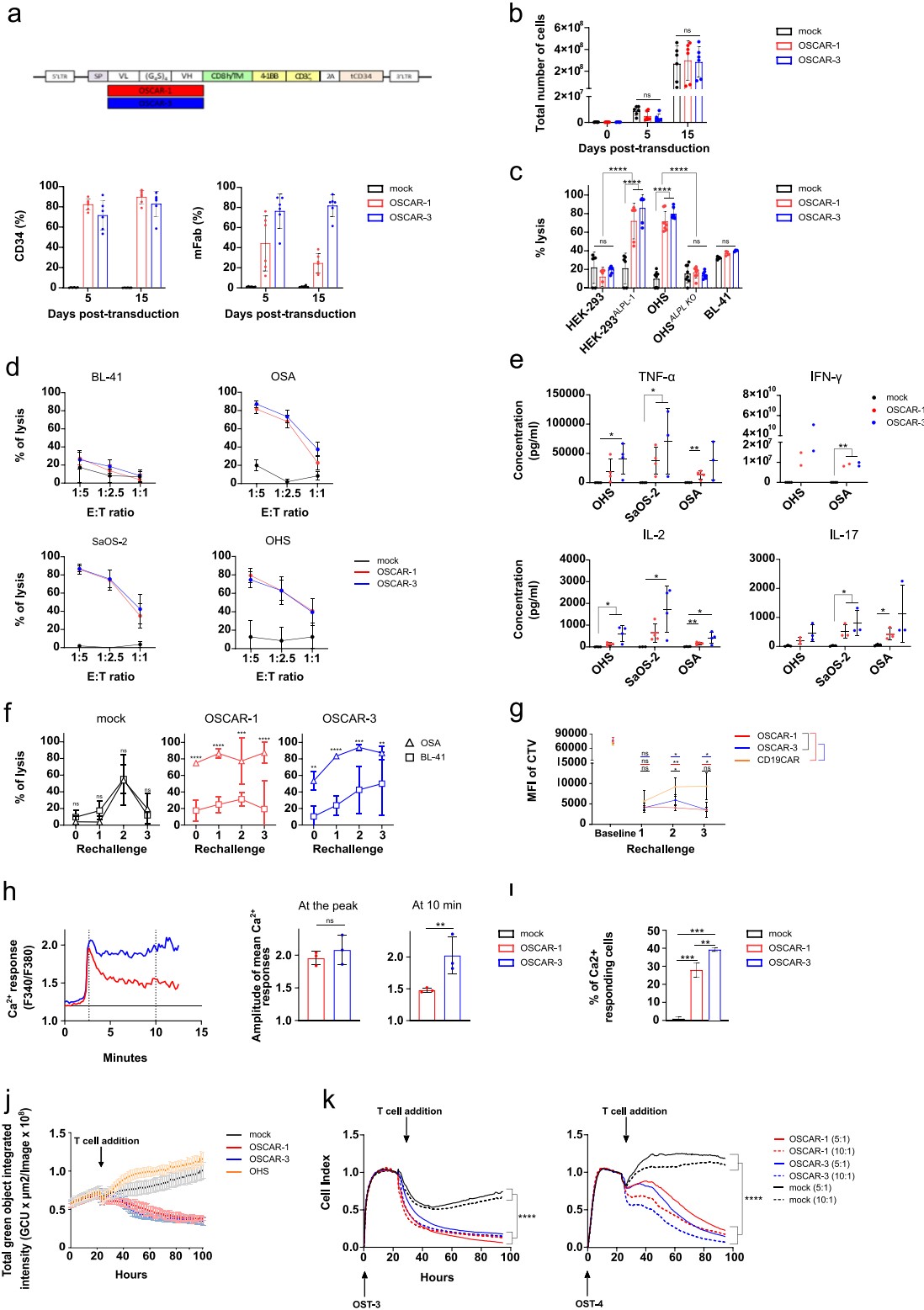

staining and observed a specific growth of OSCAR T cells on ALPL-1+ feeder cells, whereas CD19CAR T cells did not expand (Fig. 2g, Supplementary Fig. 2b). We also monitored the CAR expression upon repeated antigen exposure and observed a continuous increase of OSCAR-1 and OSCAR-3, but not CD19CAR expression (Supplementary Fig. 2c), suggesting that OSCAR T cells maintained a substantial antigen-driven proliferation capacity. These results were further corroborated when studying the intracellular $Ca^{2+}$ increase, one of the earliest signs of T lymphocyte activation, in live cell imaging[26]. Here,

OSCAR-3 T cells, in contact with OSA target cells, showed a stronger $Ca^{2+}$ response. Specifically, T cells expressing OSCAR-3 exhibited a more sustained $Ca^{2+}$ response compared to those expressing OSCAR-1 (Fig. 2h and Supplementary Movie 1, 2 and 3), indicating a stronger activation of these cells when in contact with OSA cells. In addition, this increased activation was reflected by a greater number of $Ca^{2+}$ activated OSCAR-3 T cells than OSCAR-1 T cells (Fig. 2i). Finally, we tested the capacity of OSCAR T cells to react against integrated systems. First, we generated OHS multicellular spheroids and co-cultured them with

**Fig. 2 | OSCAR T cell characterization. a** Design of OSCAR CAR construct(top). Quantification of CD34 and mFab expression of OSCAR-1 and OSCAR-3 in primary T cells (bottom). Data are presented as percentage of CD34+ and mFab+ cells ± s.d. pooled from six donors (*n* = 6). **b** Quantification of OSCAR-1 and OSCAR-3 T cell expansion. Average of the absolute cell counts taken on day 0, 5 and 10 following CD3/CD28 bead expansion. Data pooled from three independent experiments on six donors (*n* = 6). Statistical comparisons performed by two-way ANOVA with Tukey's multiple comparison test (ns not significant *p* > 0.05). **c** Cytotoxicity of OSCAR-1 and OSCAR-3 against ALPL-1 positive cells using a bioluminescence assay (BLI) measured at 6 h. CAR T cells from three donors (*n* = 3) and biological duplicates were used. Statistical comparisons performed using two-way ANOVA with Tukey's multiple comparison test (ns not significant *p* > 0.05, ****p* < 0.0001). **d** Cytotoxicity of OSCAR-1 and OSCAR-3 by BLI measured at 10 h at different E:T ratios (BL-41 negative control). Four independent experiments were pooled (*n* = 4). **e** Cytokine secretion upon overnight co-culture with ALPL positive cells. (*n* = 4 for all except IFN-γ *n* = 2). SaOS-2 IFN-γ values were out of range for all separated experiments and were removed. Statistical comparisons obtained by two-tailed unpaired Student *t*-test (ns not significant *p* > 0.05, **p* < 0.05, ***p* < 0.01). **f** Cytotoxicity of OSCAR-1 and OSCAR-3 upon repeated antigen stimulation (three). Data pooled from three donors (*n* = 3). Statistical comparisons performed with two-way ANOVA (ns not significant *p* > 0.05, **p* < 0.05, ***p* < 0.01, ****p* < 0.001, *****p* < 0.0001). **g** Proliferation of Cell Trace Violet (CTV)-labelled T cells (OSCARs,CD19CAR) upon repeated stimulation with ALPL positive targets (HEK-293$^{ALPL-1}$). Data pooled from four donors (*n* = 4). Statistical comparisons performed with two-tailed paired Student *t*-test (ns not significant *p* > 0.05, **p* < 0.05, ***p* < 0.01). **h** Ca$^{2+}$ response of OSCARs upon contact with ALPL positive cells. Ca$^{2+}$ levels are plotted against time (minutes) (left). Basal Ca$^{2+}$ levels are indicated by a black line at "1.2" on the y-axis from mock T cells. Amplitude of Ca$^{2+}$ responses (right) at the peak and at 10 min. **i** Quantification of Ca$^{2+}$ responding OSCAR-1 and OSCAR-3 T cells. **h, i**, Data pooled from three experiments (*n* = 3). From 50 to 75 cells were counted per condition. *p* values obtained by two-tailed unpaired Student *t*-test (ns not significant *p* > 0.05, ***p* < 0.01). **j** Live cytotoxicity of OSCAR-1 and OSCAR-3 T cells against OHS spheroids using Incucyte. Cytotoxicity measured as loss of green fluorescence in GFP$^+$ spheroids. Data are mean ± s.d. of sextuplicates. Data are from one representative experiment. Two independent experiments were performed (*n* = 2). **k** Cytotoxicity of OSCAR-1, OSCAR-3 on primary OS (OST-3 and OST-4) measured by xCELLigence at two E:T ratios (10:1; 5:1). Data pooled from three experiments (*n* = 3). Statistical comparisons were performed with one-way ANOVA test with Tukey's post hoc test (ns not significant *p* > 0.05, *****p* < 0.0001). For (**a–h**, and **k–l**), data are mean ± s.d.

OSCAR or mock T cells (Fig. 2j). As shown, OSCAR T cells were able to penetrate and destroy the spheroid structure. Next, we measured OSCAR-mediated cytotoxicity against two OS patient primary cells (OST-3 and OST-4) (Fig. 2k). As shown, both OSCAR constructs were able to redirect T cells against these targets and caused significant cell death of primary OS cells at different E:T ratios. From these data we concluded that OSCAR-1 and −3 recognized the same target as the TP antibodies, but with different functional avidities, and generated CARs that when expressed in T cells exhibited potent efficacy against OS targets.

## OSCAR T cells control tumor growth in human xenograft models

The efficacy of OSCAR T cells was tested in vivo. We first established a model in which green fluorescent protein (GFP) and luciferase-expressing OHS (GFP/Luc$^+$) cells were injected i.p. in mice (Fig. 3a). As shown, the tumor spread in the peritoneum, but was efficiently controlled by the two constructs (Fig. 3b, c). This was further confirmed by the significantly prolonged survival of the animals (Fig. 3d). Next, OSCAR T cells were tested in two lung metastasis models. Here, the animals were injected i.v. with GFP/Luc$^+$ OS lines known to niche in the lungs[27]; the highly metastatic SaOS-2-derivative LM7[28] (Fig. 3e–h; Supplementary Fig. 3a–c) and OSA (Supplementary Fig. 4a–e), the latter reported to be one of the most aggressive OS cell lines[27]. We first assessed in vitro OSCAR T cell reactivity against LM7 and showed that TP-1 and −3 antibodies as well as OSCAR-1 and −3 T cells could recognize this target (Supplementary Fig. 3a–c). The two OS cell lines efficiently colonized the lungs (Fig. 3f and Supplementary Fig. 4c), and although extremely aggressive, the growth of the metastases was delayed by OSCAR T cells and survival was improved accordingly (Fig. 3h and Supplementary Fig. 4d).

We analyzed the lungs from the OSA animal model and could detect T cells in the lungs of treated mice (Supplementary Fig. 4e). The capacity of OSCAR to block tumor metastasis was further assessed in an orthotopic model where OHS cells were injected in the tibia of the animals and OSCAR-1 and OSCAR-3 T cells were injected i.v. 4 days later (Fig. 4a–d). A second dose of T cells was infused on day 12. A 100-fold higher tumor growth was observed in the animals treated with mock T cells than in the OSCAR groups (Fig. 4c). Furthermore, the OSCAR T cell biodistribution was analyzed in different organs by flow cytometry. Similar levels of OSCAR-1 and OSCAR-3 T cells were found in the tibia, spleen, collateral bone marrow, and in the peripheral blood (Fig. 4d). From these data we concluded that OSCAR T cells were efficient in controlling several tumor cell lines at different anatomical locations in vivo.

## Safety assessment of OSCARs

ALPL-1 was never assigned as a target for antibody-based therapy, probably because it is found in different healthy tissues and its cellular location remains controversial being reported both at the plasma membrane and in the cytosol[29]. Membrane ALPL expression has previously been described in healthy tissues such as osteoblasts, and suspected in small bronchi epithelium of the lung, kidneys, and liver[30,31]. However, previous reports on TP antibodies only detected reactivity against osteoblasts[18]. Our present immunohistochemistry staining also confirmed that ALPL-1 expression was restricted to OS samples, and almost not detectable in healthy tissues (Fig. 1g). To assess OSCAR safety, we tested their cross-reactivity against a panel of healthy tissues in co-culture assay, followed by a detection of stimulation. We first isolated MSCs which could be differentiated into osteoblasts, and tested OSCAR T cell reactivity (Fig. 5a). Here, OSCAR T cells were incubated with MSCs or MSCs differentiated into osteoblasts for 6 and 18 days. As shown, only MSCs differentiated for 18 days were able to stimulate OSCAR-3 T cells to a similar magnitude as the control OS cells. In agreement with the lower in vitro activity of OSCAR-1, we observed slight reactivity of these cells against osteoblasts (Fig. 5a). Thus, in a clinical setting, there may be recognition of osteoblasts, in particular for OSCAR-3 T cells. We further checked for OSCAR T cell reactivity against selected primary tissues or healthy tissue cell lines, human renal epithelial cells (HREpC), human hepatocytes (HH), human pulmonary alveolar epithelial cells (HPAEpiC), human lung endothelium cells (Hulec-5a), and human fetal lung fibroblasts (MRC-5) in a co-culture assay and monitored TNFα expression in T cells. None of these tissues could induce OSCAR T cell stimulation whereas the two control OS cell lines could (Fig. 5b). We then performed membrane staining by flow cytometry using TP-3 antibody or a commercial anti-ALPL antibody of different tissues, either primary cells or cell lines representing different linages including OS as positive controls. We normalized these data based to the background and plotted the results separately (Fig. 5c). As shown, although there is some correlation between ALPL and TP-3 staining, more samples were positive for the anti-ALPL antibody and only OS cell lines were stained by TP-3 (Fig. 5c and Supplementary Fig. 5). Thus, these data indicate that plasma membrane expression of the ALPL-1 isoform is probably restricted to malignant tissues or developing bone. Finally, we assessed the safety of OSCAR T cell against BM-resident cells by co-culture, followed by colony forming unit (CFU) assay. As shown (Fig. 5d), none of the OSCARs impaired the clonogenic potential of hematopoietic stem/progenitor cells (HSPC), suggesting that OSCAR T cell cross-reactivity might be restricted to developing bone.

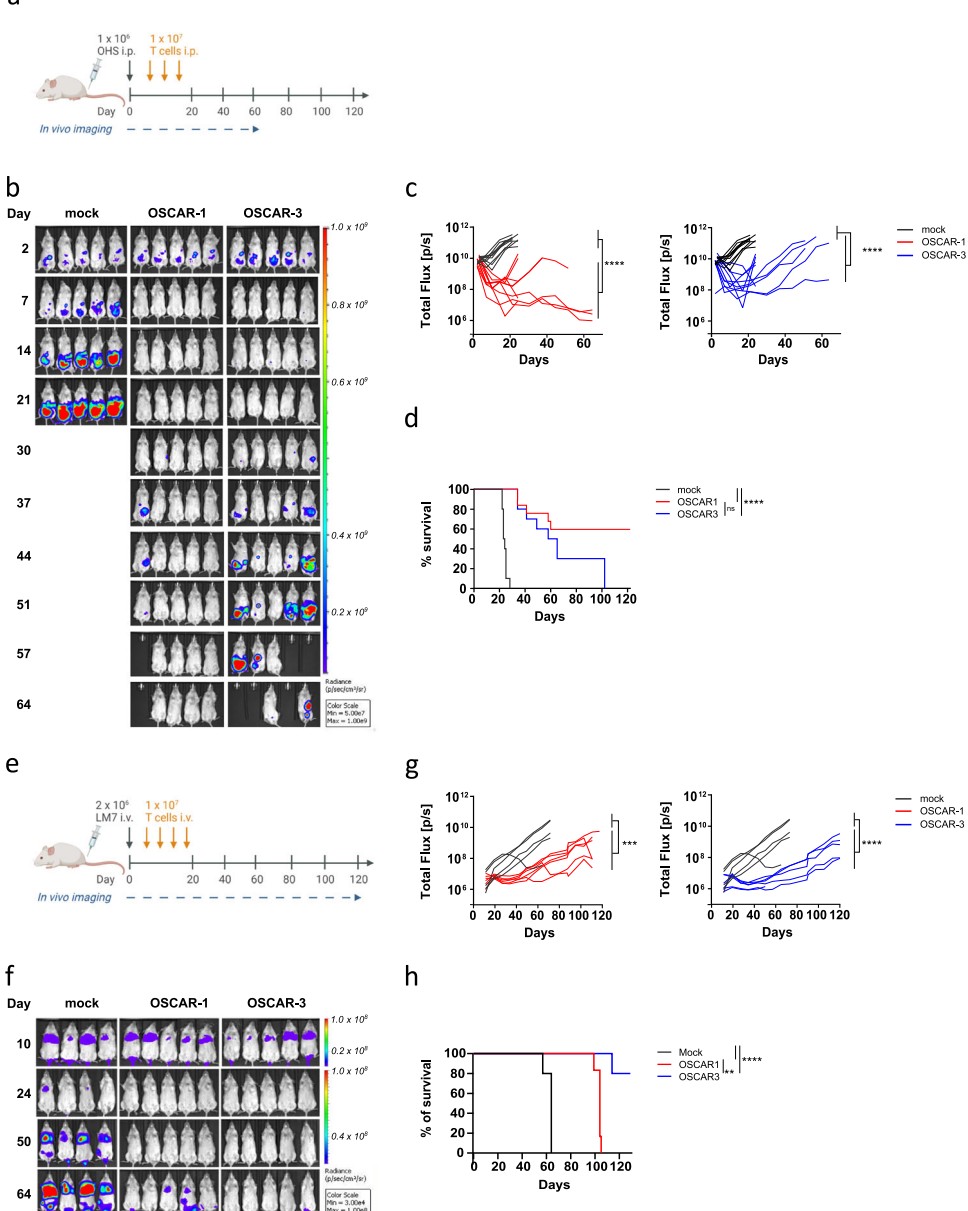

**Fig. 3 | OSCAR-1 and OSCAR-3 efficiently control tumor growth in xenograft OS models. a** Experimental overview of OHS OS model. NSG mice were engrafted with GFP/Luc$^+$ OHS cells i.p. On day 3 mice were randomized and received 3 rounds of i.p. injections of OSCAR-1, OSCAR-3, or mock transduced T cells. **b** Tumor burden measured by IVIS. Representative BLI images of the luminescent signal from each treatment group. **c** Quantification of tumor progression for each individual mouse measured by flux values acquired via BLI. OSCAR-1 vs mock T cells (left) and OSCAR-3 vs mock T cells (right). Two independent experiments of five mice per group were pooled ($n = 10$). Each line represents one individual mouse. Two-way ANOVA with Tukey's multiple comparison test was used to calculate $p$ values (ns not significant $p > 0.05$, ****$P < 0.0001$). **d** Kaplan-Meier survival curves analyzed using a Mantel-

Cox (log-rank) test (ns not significant $p > 0.05$, ****$p < 0.0001$). **e** Experimental overview of LM7 OS lung model. NSG mice were engrafted with GFP/Luc$^+$ LM7 cells i.v. On day 10 mice were randomized and received 4 i.v. injections of OSCAR-1, OSCAR-3, or mock T cells. **f** Representative BLI images of the luminescent signal are shown from each treatment group. **g** Quantification of tumor progression for each individual mouse measured by flux values acquired via BLI as in (**c**). One experiment is shown ($n = 5$ per group). Two-way ANOVA with Tukey's multiple comparison test was used to calculate $p$ values (ns not significant $p > 0.05$, ***$p < 0.001$, ****$p < 0.0001$). Representative of two separate experiments. **h** Kaplan-Meier survival curves analyzed with a Mantel-Cox (log-rank) test (ns not significant $p > 0.05$, **$p < 0.01$, ****$P < 0.0001$).

## Discussion

In the present study we identified the target of antibodies isolated more than 30 years ago, which showed great reactivity against OS and in particular against lung metastases[17,18]. These antibodies were specific for the same target, ALPL-1, among a library of 5000 transmembrane proteins. Importantly, ALPL-1 is a splice variant of the *ALPL* gene; another isoform, ALPL-3, was not recognized by TP antibodies, suggesting a restricted specificity. *ALPL-1* expression was then analyzed

using RNA-seq resources; the messenger RNA was found in different tissues, but at higher levels in OS samples. We tested the protein expression using a large panel of cell lines (flow cytometry) and healthy tissue (IHC) and confirmed the OS and osteoblast restriction of ALPL-1. Importantly, although *ALPL-1* mRNA was detected in the lung samples, the protein was not detected by TP-3 suggesting that ALPL-1 might either change location or not be transcribed in certain cell types. We then built CAR molecules based on the published sequences[23] and

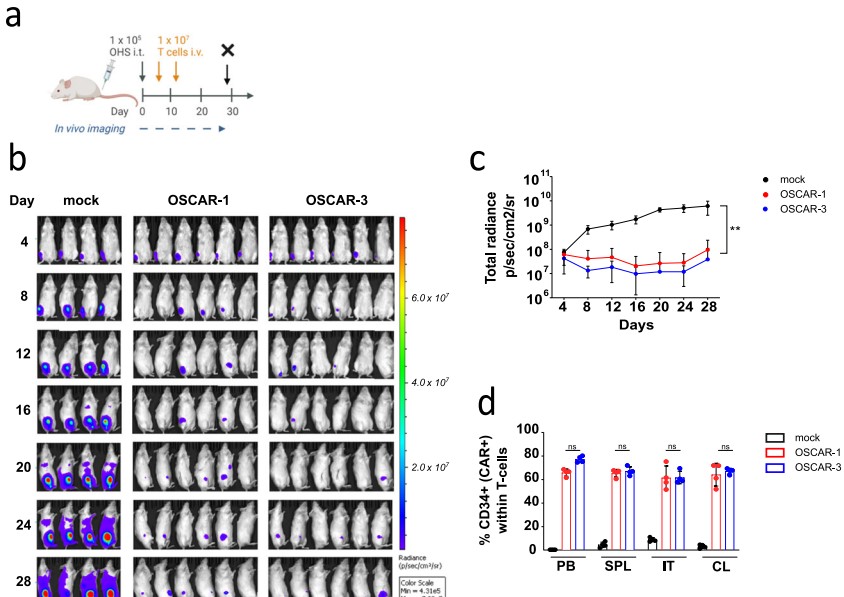

**Fig. 4 | OSCAR-1 and OSCAR-3 efficiently control tumor growth in xenograft orthotopic OS model. a** Experimental overview of OHS orthotopic OS model. NSG mice were engrafted with GFP/Luc+ OHS cells implanted within the tibiae (intra-tibial, i.t.). Mice received two doses of OSCAR-1, OSCAR-3, or mock T cells and were sacrificed at day 28 (X). **b** Tumor burden measured by IVIS. Representative BLI images of the luminescent signal from each treatment group. **c** Quantification of tumor progression as mean ± s.d. per treatment group. Statistical comparisons were performed with one-way ANOVA test with Tukey's multiple comparison test (ns not significant $p > 0.05$, **$p < 0.01$). Representative experiment is shown (OSCAR-1 $n = 6$ mice, OSCAR-3 $n = 6$ mice, mock $n = 4$ mice). **d** Detection of OSCAR-1, OSCAR-3, and mock T cells in peripheral blood (PB), spleen (SPL), intra tibia (IT) and collateral tibia (CL) from mice sacrificed on day 28 ($n = 4$ per group). Data are presented as mean ± s.d. Statistical comparisons were performed with one-way ANOVA test with Tukey's multiple comparison test (ns not significant $P > 0.05$).

demonstrated that T cells expressing these constructs were able to react against OS samples and kill them. We further performed a pre-clinical CAR T cell validation and showed that OSCAR T cells efficiently controlled tumor progression in different human xenograft models, including orthotopic and metastatic settings. Our in vivo data were run in various conditions and high doses of OSCAR T cells were used for some models, this was justified by (i) our intention to detect efficacy in a qualitative manner and (ii) have enough T cells to control the high load of tumor cells injected. It would be interesting to now run stress test in the metastatic model in order to compare the robustness of each OSCAR in a quantitative manner. It is difficult to speculate which of the two CARs is the most efficient, however, OSCAR-3 seemed more potent than OSCAR-1 in controlling the highly metastatic, lung-niching and slow growing LM7 cell line. Unexpectedly, OSCAR-1 which has a lower functional avidity in vitro mediated a superior control of fast growing tumors in vivo, and appeared more robust in a rechallenge assay which is in agreement with the recent report of Greenman and colleagues proposing that a mid-range avidity CAR might be more efficient than a higher avidity CAR for full T cell stimulation[32]. Thus OSCAR-3 could be more tonic than OSCAR-1 and become exhausted faster in antigen re-challenge experiments. Although our findings do not rule out a difference in tonicity, the difference between these constructs might be more subtle. OSCAR-1 was less detected by anti-Fab antibodies, which might be due to a lower expression of the molecule or a low detection of this polyclonal antibody which was raised against a variable protein. If the second proposition is true, some improvements in the construct such as change in the CAR scaffold (hinge, transmembrane or signaling domain) or humanization[33] could increase its expression. However, OSCAR-1 activity and its efficiency in controlling tumor suggest that its present design might be optimal. Nevertheless, the ideal CAR T cell targeting OS might require more than design modifications and rather an enhanced resistance to the strong TME.

We also tested the cross-reactivity of OSCAR T cells against healthy tissues and only detected anti-osteoblast reactivity, which

confirmed previous observations with TP antibodies. Young OS patients with open growth plates have increased osteoblastic activity. Several chemotherapeutic agents used against OS affect skeletal development and bone remodeling. Taking these clinical situations into consideration, our results support the introduction of OSCAR in clinical trials for OS treatment and a further development of anti-ALPL-1 targeted drugs.

OS treatment has not been significantly improved since the 1980's when combination chemotherapy was introduced, so there is a clear unmet clinical need for this cancer. The clinical challenge lies in controlling the hematogenous spread of malignant cells. Indeed, if the tumor colonizes the lungs and presents with overt metastases, the survival rate drops[12]. Numerous therapeutic solutions have been tested and results were mostly disappointing. Immunotherapy, which is now considered as the fourth pillar in cancer treatment, has in OS so far not been more successful[10]. Different antigens have been targeted by CAR in OS[13–15,34], the most advanced being anti-HER2 CAR, which was successfully tested in a phase I/II sarcoma trial, and showed no evidence of toxicity[14]. A challenge with this CAR is the heterogeneous HER2 expression: only 60–70% of the OS lung metastases are HER2 +, which reduces the number of eligible patients. In addition, HER2 expression levels are low in OS[35] and can be present at higher levels in healthy tissues which can lead to severe safety issues for the patient[36]. ALPL-1 has been detected in more than 90% of the OS lung metastases samples. Surprisingly, its mRNA was also found in healthy tissues, but we could neither detect the protein at the cell surface nor did we observe OSCAR T cell activation. It is tempting to speculate that the ALPL-1 cellular location varies, and that its presence at the plasma membrane might be a hallmark of cancer development, something that was already described for ER resident proteins[37]. Similarly, ALPL is depicted as both a cytosolic and membrane-bound glycosylated enzyme by the Protein Atlas database (https://www.proteinatlas.org/ENSG00000162551-ALPL), but we detect it at the membrane in OS cell lines or when overexpressed in cells. In contrast, in testing of healthy tissues, while they were

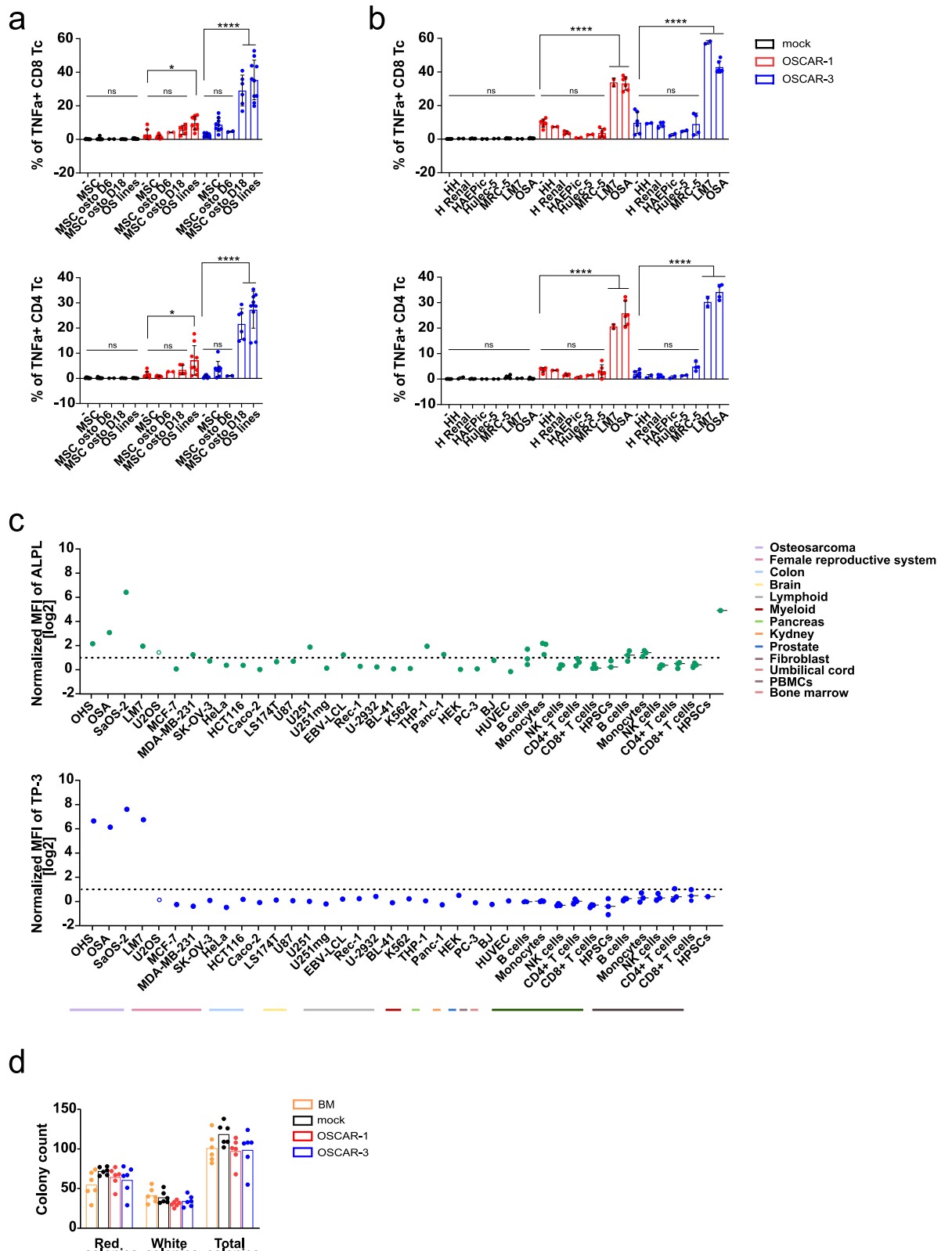

positive for the commercial anti-ALPL antibody, the TP antibody did not react. Therefore, ALPL-1 may be considered a moving target with unique TP-reactive epitopes expressed on the OS cell membrane. Similar antigens might be difficult to isolate since the concentration of mRNA does not reflect the protein levels and distribution[38], yet most of the CAR targets have been defined by transcript analysis. Thus, direct protein study following our strategy of screening cDNA

libraries, or the approach recently reported by Nix and colleagues[39] using surface proteomics to define CD72 as a target for B-ALL, should become standard for defining innovative CAR targets. Another explanation for the discrepancy between mRNA and the protein signal could come from secondary modification of ALPL-1 in OS and osteoblasts that can only be detected by TP antibodies. Although out of the scope of the present study, this work would be essential to

**Fig. 5 | OSCAR T cell safety assessments. a, b** OSCAR T cell cytotoxic activity measured as TNFα production and expression on CD8 (top) and CD4 (bottom) T cells upon co-culture with different normal tissues. TNFα expression is detected by flow cytometry intracellular staining at 6 h co-incubation at an E:T ratio of 1:2. **a** OSCAR T cell cytotoxicity against human MSCs and MSC-derived osteoblasts at different stages of differentiation. Data are mean ± s.d. pooled from three independent experiments ($n = 3$). OSCAR T cells are from three healthy donors. Human MSC are from two donors. Statistical comparisons were performed with two-way ANOVA with Tukey's multiple comparison test (ns not significant $p > 0.05$, *$p < 0.05$, ****$p < 0.0001$). **b** OSCAR-1 and OSCAR-3 T cell reactivity against healthy tissue cells representing lung (MRC-5, Hulec-5a, HPAEpiC), liver (HH), and kidney (HREpC). Data are mean ± s.d. pooled from three independent experiments ($n = 3$). OSCAR T cells are from three healthy donors. Statistical comparisons were performed with two-way ANOVA with Tukey's multiple comparison test (ns not significant $p > 0.05$, ****$p < 0.0001$). **c** ALPL and TP-3 protein expression in different tumor cell lines from osteosarcoma (OHS, OSA, SaOS-2, LM7, U2OS), other tumors (MCF-7, MDA-MB-231, SK-OV-3, HeLa, HCT116, Caco-2, LS174T, U87, U251, U251mg, EBV-LCL, Rec-1, U-2932, BL-41, K562, THP-1, PANC-1, HEK, PC-3) and cells from different healthy tissues (BJ, HUVEC), distinct cell types from PBMCs ($n = 3$), and BM ($n = 3$, except HSCs $n = 1$). Results were normalized by the ratio between the Mean Fluorescence Intensity (MFI) obtained after staining with ALPL (green) or TP-3 (blue) and that of the appropriate control, before log2 transformation. U2OS is empty circle. **d** OSCAR-1 and OSCAR-3 T cell reactivity versus hematopoietic stem and progenitor cells (HST) from healthy bone marrow by colony-forming unit (CFU) assay. OSCAR T cells and HST were co-cultured at a 10:1 E:T ratio. Data represents the number of counted colonies; white, red, and total (mixed) colonies. Data are shown as mean ± s.d. pooled from two independent experiments with two donors ($n = 2$).

further secure and eventually isolate alternative antibodies or binding proteins.

Another central question to answer in future work would be to define the role of ALPL-1 in OS and osteoblastic activity. A significant amount of literature has been published on the importance of alkaline phosphatase activity in OS[40,41] and diverse types of cancer such as prostate cancer[42], but the precise role of the isoform ALPL-1 at the membrane is not yet revealed. Interestingly, the correlation between alkaline phosphatase identity and activity was never clearly established. Considering the previous study of Mohseny and colleagues[43]; alkaline phosphatase basal activity was detected in OHS cells, whereas OSA and SaOS-2 cells were negative. However, all three cell lines were positive in our staining for ALPL-1, suggesting that ALPL-1 might not be the only phosphatase present in OHS cells, or may be differentially regulated. In addition, ALPL-1 did not seem to have a vital role since fitness of our knock out cell lines was not affected, and they grew as per the original line (our observations). However, the phenotype of the KO cells might be more subtle and require conditions closer to their real environment rather than plastic plates. Thus, future studies should also aim at describing KO cell behavior in 3D organoids or when injected into mice.

Finally, the safety of OSCAR constructs was demonstrated in co-culture assays with hematopoietic stem cells and other healthy tissues. Nevertheless, we cannot exclude the possibility that ALPL-1 isoform expression might take place under specific conditions which cannot be mimicked in vitro. Our next test of safety will be to perform a first in man clinical trial in a transient setting. We are presently testing OSCAR-3 as mRNA (Casey et al. *in preparation*) with the ambition to treat OS patients with lung metastases and unmet clinical need. The advantages of running a first in man using transient systems have been discussed, and are particularly relevant when validating new targets[44]. By combining dose escalation and careful monitoring of the patients, we believe that we will obtain crucial information on the safety of the product and hopefully confirm ALPL-1 as an attractive target for CAR-based therapy in OS.

## Methods
### Cell lines, biopsies, reagents and media
The human OS cell line, OHS, was established at the Norwegian Radium Hospital[22]. The OS cell lines OSA, U2OS, SaOS-2, 143B, G-292, and MG-63, the Burkitt Lymphoma cell line BL-41, and the human liver cancer cell line HepG2, were obtained from the American Type Culture Collection (ATCC). The human metastatic OS LM7 cell line was a kind gift from Dr. Eugenie Kleinerman (The University of Texas MD Anderson Cancer Center). Cell lines were cultured in RPMI 1640 (Thermo Fisher Scientific), supplemented with 0.05 mg/mL gentamycin (Thermo Fisher Scientific) and 10% fetal calf serum (FCS) (Thermo Fisher Scientific). Human embryonic kidney cell line HEK-293 was from our collection. HEK-293 cells were maintained in DMEM (Invitrogen, Thermo Fisher Scientific), supplemented with 10% FCS (PAA Laboratories, Fisher Scientific) and 0.05 mg/mL gentamycin. The OST-3 and OST-4 primary cell lines were generated from OS samples surgically resected at the Hospital Universitario Central de Asturias (Oviedo, Spain) after obtaining the approval of the Institutional Ethics Committee of the Principado de Asturias. OST-3 derives from a conventional osteoblastic OS resected from a 10-year-old female patient and OST-4 corresponds to a dedifferentiated OS from a 69-year-old female patient. Primary cell lines were cultured in DMEM with 10% FCS, 1% Glutamax (Thermo Fisher Scientific), and 100 U/mL penicillin/streptomycin (Thermo Fisher Scientific). Primary sarcoma tumors were collected at the Center for Orthopedic Innovations of the Mercy Miami Hospital (Florida, USA), according to regulations specified by the Nova Southeastern University Institutional Review Board (protocol # 2017-304). The human MRC-5 cell line (fibroblast from normal lung tissue) and the Hulec-5a cell line (endothelial cells from normal lung tissue) were obtained from the ATCC. MRC-5 were grown in Eagle's Minimum Essential Medium (EMEM, ATCC) supplemented with gentamicin and 10% FCS. Hulec-5a cells were cultured in MCDB131 medium (Thermo Fisher Scientific) supplemented with 10 ng/mL human epidermal growth factor (EGF; PeproTech), 1 μg/mL hydrocortisone (STEMCELL Technologies), 10 mM glutamine (Thermo Fisher Scientific) and 10% FCS. Primary human renal epithelial cells (HREpC) were purchased from PromoCell (Promocell) and maintained in ready-to-use Renal Epithelial Cell Growth Medium 2 (PromoCell). Primary human hepatocytes (HH) and primary human pulmonary alveolar epithelial cells (HPAEpiC) were purchased from ScienCell Research Laboratories (ScienCell Research Laboratories) and maintained in Hepatocyte Medium (ScienCell Research Laboratories) or Alveolar Epithelial Cell Medium (ScienCell Research Laboratories), respectively. Human primary mesenchymal stem cells (MSC) used for osteoblast differentiation experiments were isolated from healthy donor BM and maintained in MEMα supplemented with 1% GlutaMAX (Thermo Fisher Scientific) and 20% FCS. MSC osteoblast differentiation was induced using the StemPro Osteogenesis Differentiation Kit (Thermo Fisher Scientific). MSCs were also obtained from BM aspirates of healthy donors from the Catalan Blood and Tissue Bank (BST), from cord blood-derived Warton Jelly (WJ) and from fetal liver (FL), (Institutional Review Board of JCLRI approval, HCB/2017/1056). BM-MSCs, WJ-MSCs and FL-MSCs were cultured in Advanced DMEM (Thermo Fisher Scientific) supplemented with 10% heat-inactivated FBS (Thermo Fisher Scientific), 1% Glutamax (Thermo Fisher Scientific), and 100 U/mL penicillin/streptomycin (Thermo Fisher Scientific). All cell lines were passaged for less than 6 months after purchase. Cell lines were tested for mycoplasma contamination using a PCR-based detection kit (VenorGeM; Minerva Biolabs). T cells and BM were derived from healthy donors. Use of T cells and BM was approved by the Regional Committees for Medical Research Ethics South East Norway (approvals 2013/624 and 2019/121). Furthermore, the use of PBMC material was approved by the Barcelona Clinic Hospital Ethics Committee on institutional review board (IRB) (approval HCB/2018/0030), and the use of

BM sample material was approved by the Barcelona Clinic Hospital Ethics Committee (approval HCB/2017/1056). All T cells were grown in X-Vivo 15 (Lonza), supplemented with 5% CTS serum replacement (Thermo Fisher Scientific) or 5% human serum (TCS Biosciences Ltd) and 100 U/mL interleukin (IL)−2 (Proleukin; Novartis), referred to as complete medium hereafter, unless otherwise stated.

## Osteosarcoma patient-derived xenografts (PDXs)

We obtained viable OS PDX tumors established from seven patients at Hospital Sant Joan de Déu (HSJD, Barcelona). Clinical details of these PDX have been published elsewhere[45]. Briefly, freshly excised, or cryopreserved PDX fragments were mechanically and enzymatically dissociated to obtain tumor single-cell suspensions. Small 2–4 mm³ fragments were digested with 200 U/mL Collagenase Type IV (Thermo Fisher Scientific) for 20 min at 37 °C. Afterwards, disaggregated tissues were washed with PBS and treated with Cell Dissociation Buffer (Thermo Fisher Scientific) for 20 min at 37 °C. Single-cell suspensions were filtered through a 100-μm cell strainer for immediate used.

## Immunohistochemistry (IHC)

Four-micrometer-thick tissue sections from frozen samples were fixed with 10% formaldehyde during 10 min, immersed in 3% H2O2 aqueous solution for 30 min to exhaust endogenous peroxidase activity and then covered with 1% blocking reagent (Roche) to block nonspecific binding sites. Sections were incubated with primary antibody TP-3 during 7 min and then peroxidase-labeled secondary antibodies and 3,3-diaminobenzidine were applied to develop immunoreactivity, according to manufacturer's protocol (BOND Polymer Refine Detection; Leica Biosystems). Finally, slides were counterstained with hematoxylin and mounted in DPX (BDH Laboratories). The use of normal and osteosarcoma tissues for IHC staining was approved by CEI de los Hospitales Universitarios Virgen Macarena y Virgen del Rocío (v.1 07/12/2021).

## DNA/RNA constructs

OSCAR constructs were derived from the published sequences[23] and the corrected sequences published with the GenBank numbers: AJ131748.1 (TP-1 heavy chain), AJ131747.1 (TP-1 light chain), AJ131750.1 (TP-3 heavy chain), and AJ131749.1 (TP-3 light chain). An scFv was designed including a $(G_4S)_4$ linker between VL and VH regions, and codon-optimized sequences were ordered (Eurofins MWG). These scFvs were cloned into a second generation CAR vector containing a CD8 hinge and transmembrane domain fused to 4-1BB and CD3ζ[24]. All constructs were initially cloned into a pENTR vector and later recombined into compatible expression vectors by Gateway cloning (Themo Fisher Scientific)[46]. For the knock out study, we ordered the following sgRNA sequences: ALPL-1 5′ GGCATGGTTCACTCTCGTGG 3′, ALPL-2 5′ GATGACATTCTTAGCCACGT 3′, and ALPL-3 5′ GTTGCACCGGGA ACGCTCAG 3′ (Eurofins MWG). The retroviral plasmid encoding for Cas9, MSCV_Cas9_puro, was a gift from Christopher Vakoc (Addgene plasmid # 65655)[47]. The Luciferase-GFP construct was a kind gift from R. Löw (Eufets AG) and was used to generate luciferase-expressing target cells (GFP/Luc⁺). The vector encoding the full length human transcript variant 1 of ALPL (ALPL-1, NM_000478.4) coding sequence was clone # HG10440-UT from Sino Biological.

## Generation of ALPL knockout OHS cell line

OHS cell line was retrovirally transduced with MSCV_Cas9_puro and cultured in culture media containing 0.5 μg/mL Puromycin (Thermo Fisher Scientifics). Following antibiotic selection, the OHS-Cas9 cell line was electroporated with an ALPL-targeting sgRNA using a BTX 830 Square Wave Electroporation System (BTX Technologies). Electroporation was performed with $2 \times 10^6$ OHS-Cas9 cells and 400 pmol ALPL sgRNA in a 1-mm gap cuvette at 125 V for 2 milliseconds. Cells

were transferred to the culture media immediately after. ALPL knockout cells were sorted after staining with anti-ALPL antibody, and this lineage was used for subsequent experiments.

## Generation of ALPL expressing HEK-293 cell line

HEK-293 cells were seeded in a 60-mm cell culture treated dish at $1 \times 10^6$ cells in 3 mL of DMEM (Invitrogen, Thermo Fisher Scientific), supplemented with 10% FCS (PAA Laboratories, Fisher Scientific) and 0.05 mg/mL gentamycin. The following day, 3 μg of ALPL expression vector was complexed with 9 μl of X-tremeGENE™ 9 (Merck) in 300 μl of Opti-MEM media (Thermo Fisher Scientific) and applied to the cells. The next day, ALPL expression was confirmed by flow cytometry.

## T cell transduction and expansion

Peripheral blood mononuclear cells (PBMC) from healthy donors were isolated by density gradient centrifugation from buffy coats obtained from the Oslo blood bank (Regional Committee for Medical Research Ethics, authorization 2019/121). Retroviral transduction of PBMCs was performed by spinoculation as previously described[46]. Briefly, PBMCs were incubated for 2 days in a 24-well plate coated with anti-CD3 (1 μg/mL; OKT-3, Thermo Fisher Scientific) and anti-CD28 (1 μg/mL; CD28.6, Thermo Fisher Scientific) antibodies at $1 \times 10^6$ cells/mL. Stimulated T cells were spinoculated twice with retroviral CAR supernatants using 1 mL of virus solution deposited in each well and 500 μL of activated T cells at a concentration of $0.3 \times 10^6$ cells/mL in a non-treated 24-well culture plate (Nunc A/S) pre-coated with retronectin (50 μg/mL, Takara Bio. Inc.). T cells were then harvested and resuspended in complete medium. Transduction efficiency was checked after 3 to 7 days by flow cytometry. CAR transduced T cells were expanded for 10 days using CD3/CD28 Dynabeads (Thermo Fisher Scientific) as previously described[48].

For experiments tracking T cell expansion $0.2 \times 10^5$ T cells were spinoculated twice with retroviral CAR supernatants (day 0). T cells were then harvested and resuspended in complete medium. On day 6 transduction efficiency was checked and T cells were counted prior to initiating expansion with CD3/CD28 Dynabeads. T cells were then expanded in the presence of CD3/CD28 beads at 1 T cell: 1 bead ratio over a period of 10 days[49]. T cells were counted again at the end of the expansion (day 16).

Transduction of tumor cell lines with Luciferase-GFP construct was performed using the same protocol described above, without the pre-stimulation step.

## In vitro functional assay, antibodies, and flow cytometry

To measure TNFα production in response to target cell stimulation, CAR T cells were incubated with the following cell lines, OSA, LM7, MRC-5 and Hulec-5a, and primary human cells, HREpC, HPAEpiC, HH, MSCs and MSC-derived osteoblasts. Cells were incubated at 37 °C in 5% CO2 for 6 h at an effector cell to target cell (E:T) ratio of 1:2, in the presence of BD GolgiPlug and BD Golgistop (BD Biosciences) at 1/1000 dilution. Cells were washed twice and stained extracellularly and intracellularly using the PerFix-nc kit according to the manufacturer's instructions (Beckman Coulter). Antibodies specific for human CD4 (Clone OKT4; Biolegend; 2 μl) and CD8 (Clone RPA-T8; Invitrogen, Thermo Fisher Scientific; 2 μl) were used to identify CD4 and CD8 CAR T cells respectively. Anti-human TNFα antibody (Clone Mab11; BD Pharmigen, BD Biosciences; 10 μl) was used to measure effector functions.

To assess ALPL-1 expression on PBMCs and BM we stained cells with a panel of antibodies. Anti-CD3 (Clone SK7; BD Biosciences; dilution 1:100), CD4 (Clone RPA-T4; BD Horizon; dilution 1:100), CD8 (Clone SK1; BD Horizon; dilution 1:100), CD7 (Clone M-T701; BD Pharmigen; dilution 1:100), CD19 (Clone SJ25C1; BD Pharmigen; dilution 1:100), CD33 (Clone WM53; BD Pharmigen; dilution 1:100), and CD34 (Clone 8G12; BD Biosciences; dilution 1:100) antibodies were

used to identify the distinct blood cell populations. For PBMCs, T cells were defined as CD3 +, T helper as CD3 +/CD4 +, T cytotoxic as CD3 +/CD8 +, B cells as CD19 +, monocytes as CD33 +, NK cells as CD7 +/CD3- and HPSCs as CD34 +. BM derived mononuclear cells were labelled with CD34 magnetic beads and loaded onto an Auto-MACS ProSeparator (Miltenyi) where CD34⁺ labelled cells were efficiently separated from the CD34⁻ portion.

Tumor cell lines and cells from normal human tissues were stained with either commercial mouse anti-human alkaline phosphatase antibody (Clone B4–78; BD Pharmingen, BD Biosciences; 3 µl), anti-human alkaline phosphatase antibody (Clone B4–78; R&D; 3 µl), or with TP-1 and TP-3 antibodies from hybridomas (our collection; dilution 1:200). OSCAR expression was detected by staining T cells with either anti-human CD34 antibody (Clone 4H11; eBiociences, Thermo Fisher Scientific; 2.5 µl) or anti-mouse Fab biotinylated antibody (Jackson ImmunoResearch; dilution 1:200), and streptavidin-PE (BD Biosciences; dilution 1:400).

For extracellular staining, single-cell suspensions were labeled with the appropriate monoclonal antibodies for 15 min at room temperature (RT) in the dark, in PBS containing 2% FCS. For indirect staining, primary antibody labeling was followed by staining with a compatible fluorescent dye-conjugated secondary antibody. Goat anti-mouse IgG antibody (Clone Poly4053; Biolegend; dilution 1:200), either PE or APC conjugated, goat anti-mouse IgG1 PE (Clone 11711; R&D; dilution 1:200), or goat anti-mouse IgG F(ab')2 Alex Fluor-647 conjugated (Cell Signaling Technology; dilution 1:500) were used as a secondary antibody. Cell viability of PDXs was measured by using the 7-Aminoactinomycin D (7-AAD) (BD Via-Probe, BD Biosciences; 100 µl/test) dye. Flow cytometry was performed on a BD FACSCanto II (BD Biosciences) and data were collected with BD FACSDiva v8.0.1. Data were analyzed with FlowJo software v10.8.0 (TreeStar). All gating strategies are shown in Supplementary information file (Supplementary method 2 a–e). FACS Aria (BD Biosciences) or Sony SH800 Cell Sorters (Sony Biotechnology Inc.) were used for cell sorting.

### Antigen density test
TP-3 antigen density was measured on few OS cell lines (OHS, OSA, U2OS) and other tumor cells (BL-41, HepG2) and healthy cells with different tissue origins (MRC-5, Hulec-5, HREpC) using QIFIKIT (Dako) according to manufacturer's instructions. TP-3 antigen density was calculated and plotted as Specific Antibody-Binding Capacity (SABC).

### In vitro cytokine release assay
CAR T cells were co-cultured for 24 h with OS cell lines at an E:T ratio of 1:2. Culture supernatant was harvested, diluted 1:2, and the cytokine release was measured using the BioplexPro™ Human Cytokine 27-plex Assay (Bio-Rad Laboratories Inc.) according to manufacturer's instructions. Secreted cytokines levels were quantified using the Bio-Plex 200 system instrument (Bio-Rad Laboratories Inc) and cytokine concentration was calculated from the standard curve. Data were collected and analyzed with Bio-Plex Manager™ software version 6.1.

### In vitro bioluminescence-based CAR T cell killing assay
Green fluorescent protein (GFP) and luciferase-expressing tumor cells (GFP/Luc⁺) were counted and resuspended at a concentration of $3 \times 10^5$ cells/mL. Tumor cells were given D-Luciferin at 75 µg/mL final concentration (Perkin Elmer) and were placed in 96-well white flat-bottomed plates at 100 µL/well, in triplicate. CAR T cells were added to the wells at 10:1 E:T ratio. Target cells incubated without CAR T cells or in the presence of 1% Triton™ X-100 (Sigma-Aldrich) were used to detect spontaneous and maximal killing, respectively. Cells were incubated at 37 °C and the bioluminescence (BLI) was measured as relative light units (RLU) at indicated timepoints with a luminometer (VICTOR Multilabel Plate Reader, Perkin Elmer) and WorkOut v2.5 software (Perkin Elmer). Lysis percentage was calculated using the

following equation: % specific lysis = 100 x (spontaneous cell death RLU – sample RLU) / (spontaneous death RLU – maximal killing RLU).

### In vitro cytotoxicity of OSCAR T cells against OS primary cells
Cell death was measured using the xCELLigence RTCA DP system (Agilent), RTCA v2.1 software, which measures electrical impedance and converts impedance values into a cell index (CI). CI values increase progressively and proportionally as cells attach and proliferate and they decrease as cells detach and die[50]. Primary OS cells, OST-3 and OST-4, were resuspended in DMEM high glucose containing 10% FCS, seeded in xCELLigence plate wells at $1 \times 10^4$ cells and incubated for 24 h. OSCAR-1, OSCAR-3, and mock T cells were added after 24 h at 10:1 and 5:1 E:T ratios and the immune cell killing was monitored for 72 h.

### Rechallenge assay
CAR T cells were co-cultured with irradiated OHS cells expressing ALPL in complete T cell culture media, at an E:T ratio of 1:2 ($1 \times 10^6$ of T cells for $2 \times 10^6$ of OHS) for 7 days. On day 7, T cells were harvested, counted and co-incubated with irradiated OHS cells, under the same experimental conditions, for 7 days (2nd rechallenge). This was repeated three times for a total of three rechallenges. At the end of each rechallenge CAR T cells were checked for CAR expression and cytotoxicity (BLI). In the BLI assay, CAR T cells were co-cultured with an OS cell line positive for ALPL (OSA) but different from the OS cells use for the rechallenge co-culture. BL-41 cells were used as ALPL negative target cells.

### CTV-based proliferation assay
CAR T cells were resuspended at $1 \times 10^6$ cells/ml, twice washed with 1× PBS and stained using CellTrace® Violet (CTV, Life Technologies) at a final concentration of 2.5 µM for 20 min at room temperature, protected from light. The reaction was quenched by adding 5 times the original staining volume of X-vivo 15, 5% human serum and incubated for 10 min in the dark. Cells were pellet by centrifugation and resuspended in complete culture medium. Then, CTV-labelled T cells were plated in 24-well plate in the presence of irradiated HEK-293 cells ectopically expressing (or not) ALPL1 used as stimulator at an E:T ratio of 1:2 ($1 \times 10^6$ of T cells for $2 \times 10^6$ of HEK-293 at $1 \times 10^6$). T cells were subsequently incubated at 37 °C, 5% $CO_2$, 95% humidity for 5 days. At the end of the 5 day incubation, proliferation was assessed by flow cytometry. To measure proliferation capacity after repeated antigen exposures, we kept a proportion of T cells separate in culture and stained them with CTV on day 5 after initial stimulation, just before setting the second co-culture (2nd rechallenge). This was repeated for a total of three stimulations (rechallenges). Cells were also stained with antibodies for CAR detection, either anti-CD34 or anti-Fab.

### Colony-forming unit assay (CFU)
BM progenitor cells were pre-incubated with autologous T cells for 6 h before the cells were plated in semisolid methylcellulose medium, MethoCult™ (StemCell Technologies Inc.). CFUs were then counted and scored after 14 days following standard procedures[51].

### OS multicellular spheroids
The liquid overlay method was used to generate spheroids. Flat bottom 96-well plates were coated with 1.5% agarose (Sigma-Aldrich). GFP/Luc⁺ OHS cells were added to each well at 500 cells in 100 µL medium. The plate was centrifuged at $470 \times g$ for 15 min, and then incubated for 5 days. On day 5, $2 \times 10^5$ OSCAR-1, OSCAR-3 or mock T cells were added in 100 µL to the spheroid cultures and the plate was then put into IncuCyte S3 (Sartorius Lab Instruments) with the following settings: 1 image/hour, 2 images/well, 2 channels (phase and green). Data were collected with IncuCyte S3 2019A software. Metrics were extracted and analyzed in GraphPad software. The killing of

target cells was monitored by following the total green object integrated intensity (GCUx µm²/image) over time (100 h).

## Western blot

For lysate preparation, the cells were washed with ice-cold PBS solution and lysed in RIPA lysis buffer (Thermo Fisher Scientific) containing protease inhibitor cocktail (Thermo Fisher Scientific). The supernatant was collected after centrifugation of the cell lysate at $12000 \times g$ for 20 min at 4 °C. Samples were separated on a 4–20% Mini-PROTEAN® TGX™ gel (Bio-Rad Laboratories) and transferred to a polyvinylidene difluoride (PVDF) membrane (Bio-Rad Laboratories). The blocking was performed overnight in 5% nonfat dried milk (Sigma-Aldrich) in PBS containing 0.1% Tween 20 (Sigma-Aldrich). Membranes were then incubated under constant agitation with the following primary antibodies: human alkaline phosphatase (ALPL) antibody (Clone 928929, R&D Systems; dilution 0.25 µg/ml), IGSF-1 polyclonal antibody (ThermoFisher Scientific; dilution 1:1000), and beta actin monoclonal antibody (Clone 15G5A11/E2, ThermoFisher Scientific; dilution 1:2000), for 2 h at RT. Antibodies were used at concentrations as advised by the manufacturer. After several washes, the membrane was incubated with goat anti-mouse IgG (H + L) HRP secondary antibody (polyclonal, Thermo Fisher Scientific; dilution of 1:2500) and goat anti-rabbit IgG (H + L) HRP secondary antibody (polyclonal, Thermo Fisher Scientific; dilution of 1:2000), at RT for 90 min. Finally, the protein band intensity was visualized using Supersignal West Dura (Thermo Fisher Scientific).

## Gene expression analysis

*ALPL-1* expression was analyzed in samples of primary and metastatic OS, in various sarcomas, and in healthy lung, bone, liver, and kidney. FastQC files were downloaded from the European Nucleotide Archive, ENA (https://www.ebi.ac.uk/ena/browser/home); see supplementary data table for more details. In addition to the ENA data, sample files for osteosarcomas, synovial sarcoma, chondrosarcoma, Ewings sarcomas, extraosseous-osteosarcoma, myxofibrosarcoma, pleaomorphic spindle cell sarcoma, high grade pleomorphic fibrosarcoma, soft tissue sarcoma, fibrosarcoma, and leiomyosarcoma were generated from primary human sarcoma explants as described in[52]. Briefly, sarcoma tumors were processed within 12 h of surgical excision, then homogenous cell suspensions were serially passaged to generate primary sarcoma explants (minimum 12 passages). For RNA isolation from the primary sarcoma explants, 1 million cells with over 80% viability were lysed in RLT buffer for downstream paired-end RNA sequencing analysis. Samples were submitted for Illumina TruSeq Stranded total RNA Gold library generation and sequenced on a 2 × 150 bp paired-end run using the NextSeq 500 high Output Kit 9300-cycle; 400 million read flow cell following manufacturer's instructions. Quality control assessment was done using the Illumina RNA-seq pipeline to estimate genomic coverage, percent alignment, and nucleotide quality. Raw sequencing data were transformed to fastQ format and further analysed. Prior to analysis, the FastQ files were checked for quality issues using Trim Galore software[53] (https://www.bioinformatics.babraham.ac.uk/projects/trim_galore/) and Cutadapt[54] and FastQC (http://www.bioinformatics.babraham.ac.uk/projects/fastqc), then summarized using MultiQC software[55]. Kallisto[56] was used for quantifying abundance of transcripts. The reference transcriptome was Ensemble, GRCh38 (ftp://ftp.ensembl.org/pub/release100/fasta/homo_sapiens/cdna/Homo_sapiens.GRCh38.cdna.all.fa.gz). Prior to analysis, the transcript-level abundance estimates were imported and summarized with the tximport R-package[57] R-CRAN v4.0 (Mac). To directly relate expression for "sample-to-sample" comparisons, estimated counts from Kallisto were imported into DESeq2 v1.21.1 Bioconductor 3.11[58] and the edgeR packages v3.30.3 Bioconductor 3.11[59]. Counts were normalized by Trimmed mean of M values (TMM) method in edgeR using all available genes or transcripts. Log2 or variance stabilizing transformation (VST) were used for graphic presentations of transformed TMM normalized data. Statistical analyses comparing groups were performed using Man-Whitney test.

## Mouse xenograft studies

All animal experiments were approved by the Norwegian Food Safety Authority (approval ID11118). Non-obese diabetic (NOD).Cg-Prkdcscid Il2rgtm1Wjl/SzJ (NSG) female mice were bred in-house and maintained in a strictly controlled hygienic and pathogen-free facility under an approved institutional animal care protocol. Mice were housed in 12 h light-dark cycles with humidity between 30–70% at ambient temperature of 20–26 degrees Celsius. The study and control animals were housed in the same room. Euthanasia was conducted via exsanguination under isoflurane anesthesia followed by cervical dislocation or by cervical dislocation only. Health status of mice was monitored and mice were individually scored weekly for clinical parameters (posture, activity, fur, skin, and weight loss) to document general health condition. Humane endpoint criteria for non-subcutaneous tumor models were (i), weight loss greater than or equal to 20% from baseline, (ii), abnormal gait, paralysis, or inability to ambulate properly, (iii), respiratory distress, (iv), lethargy or persistent recumbency, and (v), abnormal neurological behaviors. Five mice (female, six-to-eight-week-old) per condition were injected intraperitoneally (i.p.) or intravenously (i.v.) with $1 \times 10^6$ GFP/Luc⁺ OS cell lines; OHS (i.p.), OSA (i.v.) or LM7 (i.v.). Tumor growth was monitored by BLI imaging with an in vivo imaging system (IVIS, Lumina III; Perkin-Elmer). Images were analyzed using the Xenogen Spectrum system and Living Image version (v.) 3.2 software (LI-COR Biosciences). Tumor-bearing mice were injected 3 or 4 times with $1 \times 10^7$ OSCAR-1 or OSCAR-3 transduced T cells, or mock T cells. T cells were injected following the same route as the tumor cells. Tumor growth was monitored weekly. Upon reaching humane endpoints, mice were euthanized.

To study T cell migration to the lung, 3 mice per treatment group were injected i.v. with $1 \times 10^6$ tumor cells, either OSA or LM7. Tumor-bearing mice were randomized and injected once with $1 \times 10^7$ OSCAR-1 or OSCAR-3 transduced T cells or mock T cells. Mice were euthanized 4 days after T cell injection and lungs were dissected. Single cell suspensions from lungs were washed repeatedly, blocked with 1 mg/mL of gamma-globulin (Sigma-Aldrich) for 10 min at RT and stained with anti-human CD3 (Clone OKT3; Invitrogen; 2 µl) and anti-human CD45 (Clone HI30, BD Biosciences; 2 µl). Cells were acquired on a BD FACSCanto flow cytometer (BD Biosciences).

For the orthotopic model seven-to-ten-week-old NSG female mice were bred and housed under pathogen-free conditions at the Animal Facility of the Barcelona Biomedical Research Park (PRBB). Animal experiments were approved by the the Spanish local ethics committee on animal experimentation and welfare (FGA-17-0030). NSG mice were injected intra-tibial (i.t) with $1 \times 10^5$ GFP/Luc⁺ OHS cells (day 0). OSCAR-1, OSCAR-3 or mock T cells were infused i.v. on day 4 ($1 \times 10^7$ T cells). Mice received a second dose of T cells ($1 \times 10^7$) on day 12 and were followed every 4 days up to day 28 by BLI imaging with IVIS. Mice were euthanized upon reaching humane endpoints. At day 28, peripheral blood (PB), spleen (SPL), injected-tibia (IT) and collateral tibia (CL) were collected from the mice, harvested and processed by FACS analysis to assess the presence of OSCAR T cells[60,61].

## Ca²⁺ flux analysis

CAR T cells and mock T cells were incubated for 30 min at 37 °C with 1 µM Fura-2 AM (Life Technologies). After washing in HBSS, CAR T cells were resuspended in TexMACS complete medium with 3% AB serum (Sigma-Aldrich). $1 \times 10^5$ CAR T cells loaded with Fura-2 were then added to a tumor cell layer (OSA) cultured in µ ibidi plates (ibidi GmbH). Images were acquired every 20 seconds with MetaFluor software v? (MetaFluor Fluorescence Ratio Imaging Software, Molecular Devices) and analysed with Image J v1.53e. T cells loaded with Fura-2 AM were alternatively excited at 350 and 380 nm. Intracellular Ca²⁺ values were

represented as a ratio: fluorescence intensity at 350 nm/fluorescence intensity at 380 nm. CAR T cells were considered responsive when the amplitude of their responses reached at least twice that of the background.

## Statistical analysis

Data visualization and statistical analysis were performed on GraphPad Prism 9 software. All experimental data are represented as mean ± s.d. Appropriate statistical tests were used to analyze data and are described in the figure legends. Statistical comparisons were made with either multi-variate bidirectional Student $t$-test (single comparison), and one-way or two-way ANOVA with multiple comparison correction when comparing more than two groups, both with Tukey correction. For in vivo studies, survival curves were compared with the log-rank estimator Mantel-Haenszel test. For gene expression data, statistical analyses comparing groups were performed using Man-Whitney test.

## Reporting summary

Further information on research design is available in the Nature Portfolio Reporting Summary linked to this article.

## Data availability

Source data are provided with this paper. ALPL-1 expression data FastQC files were downloaded from the European Nucleotide Archive, ENA (https://www.ebi.ac.uk/ena/browser/home). The reference transcriptome was Ensemble, GRCh38 (ftp://ftp.ensembl.org/pub/release100/fasta/homo_sapiens/cdna/Homo_sapiens.GRCh38.cdna.all.fa.gz). Source data are provided with this paper.

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

## Acknowledgements

The authors would like to thank our colleagues from the Translational Research Unit and the Flow cytometry Core facility of OUS for providing technical assistance. We are grateful to Gibco and Life Technologies AS for supplying CTS™ Dynabeads™ CD3/CD28 and Drs. Mengyu Wang and Hanne B. Scholz (Oslo University Hospital) for providing the mesenchymal stem cells. We thank Prof. Michael Nishimura (Loyola University Chicago Stritch School of Medicine, USA) for sharing the truncated CD34 sequence. This study was supported by the Norwegian Research Council (Grant numbers: 284983 and 316407 to S.W. and 326811 to E.M.I), the Norwegian Health Region South East (Grant numbers: 2020601, 2018579, 2016006 and 2019062 to S.W. and 2019004 to E.M.I.) and S.J. is supported by the Norwegian Research Council under the frame of the Era-Net EURONANOMED-3 European Research project "NAN-4-TUM". We thank Nova Southeastern University Center for Collaborative Research Core Facilities personnel Dr. Robin Krueger, Solly-Ann Barton-Case and Dr. Bojie Dai for their support in generation of RNAseq data. A.-M.G. was supported by the Swedish Society for Medical Research (SSMF). Research in P.M.'s Laboratory was funded by "la Caixa" Foundation Validate Program, ISCIII-RICORS within the Next Generation EU program (plan de recuperación, transformación y resilencia), and core support from CERCA/Generalitat de Catalunya and Fundació Josep Carreras-Obra Social la Caixa, the CaixaImpulse Grant CI21-00189, which has received funding from the European Institute of Innovation and Technology (EIT). This body of the European Union receives support from the European Union's Horizon 2021 research and innovation program. C.P. is supported by a PFIS fellowship from Instituto de Salud Carlos III (ISCIII) (FI21/00161). Figures 3a, 3e, 4a, ad Supplementary Fig. 4a were prepared using BioRender.com.

## Author contributions

N.M., H.K., S.J., N.P.C., P.R., C.P., R.R., L.V., A.J., M.R.M., A.F., C.C.R., S.M.M., A.D.D., A.M.G., I.G., A.D.M., A.M.C., E.A., E.D., P.M., E.M.I, and S.W. designed and performed the experiments. P.W. performed bioinformatic analyses. S.W., E.M.I., G.G., G.K., and Ø.B. conceptualized the project. S.W. and N.M. wrote the original draft, S.J., N.M., E.M.I, and S.W. prepared the figures. All authors reviewed and edited the article. S.W. and E.M.I. supervised the study. S.W., E.M.I., Ø.B., and P.M. acquired funding.

## Competing interests

S.W., E.M.I., and Ø.B. are inventors of the patent WO2020127734. A.M.G. and A.D.D. were not employees of Glycostem Therapeutics B.V. when their contribution for this study was performed. I.G. was not employee of Thermo Fisher Scientific when is contribution for this study was performed. P.M. is cofounder on OneChain Immunotherapeutics, a Josep Carreras Leukemia Research Institute spin-off company. The remaining authors declare no competing interests.

## Additional information

[1]Translational Research Unit, Department of Cellular Therapy, Oslo University Hospital, Oslo, Norway. [2]Josep Carreras Leukemia Research Institute, Barcelona, Spain. [3]Red Española de Terapias Avanzadas (TERAV)-Instituto de Salud Carlos III (ISCIII) (RICORS, RD21/0017/0029), Madrid, Spain. [4]Instituto de Investigación Sanitaria del Principado de Asturias (ISPA), Hospital Universitario Central de Asturias, Oviedo, Spain. [5]Instituto Universitario de Oncología del Principado de Asturias, Oviedo, Spain. [6]Centro de Investigación Biomédica en Red-Oncología (CIBER-ONC), Instituto de Salud Carlos III, Madrid, Spain. [7]Université de Paris, Institut Cochin, INSERM, CNRS, Equipe labellisée Ligue Contre le Cancer, F-75014 PARIS, France. [8]Department of Radiation Biology, Institute for Cancer Research, Oslo University Hospital, Oslo, Norway. [9]Institute of Biomedicine of Sevilla (IBiS), Virgen del Rocio University Hospital, CSIC, University of Sevilla, CIBER-ONC, 41013 Seville, Spain. [10]NSU Cell Therapy Institute, Dr. Kiran C. Patel College of Allopathic Medicine, Nova Southeastern University, Fort Lauderdale, FL, USA. [11]Department of Oncology-Pathology, Karolinska Institutet, Stockholm, Sweden. [12]Laboratory of Animal Cell Physiology, Graduate School of Bioagricultural Sciences, Nagoya University, Nagoya, Japan. [13]Department of Cancer Immunology, Oslo University Hospital, Oslo, Norway. [14]SJD Pediatric Cancer Center Barcelona, Institut de Recerca Sant Joan de Deu, Barcelona 08950, Spain. [15]Department of Normal and Pathological Cytology and Histology, School of Medicine, University of Seville, 41009 Seville, Spain. [16]Department of Oncology, Oslo University Hospital and Institute of Clinical Medicine, University of Oslo, Oslo, Norway. [17]CIBER-ONC, ISCIII, Barcelona, Spain. [18]Institució Catalana de Recerca i Estudis Avançats (ICREA), Barcelona, Spain. [19]Department of Biomedicine, School of Medicine, University of Barcelona, Barcelona, Spain. [20]These authors contributed equally: Hakan Köksal, Sandy Joaquina. ✉e-mail: elsin@rr-research.no; sebastw@rr-research.no

