## [Peer Review File · Nature Communications]

ALPL-1 is a target for chimeric antigen receptor therapy in osteosarcomaREVIEWER COMMENTS

Reviewer #1 (Remarks to the Author):

This is a well conducted and written study.

It is methodologically sound and all the experiments have been conducted correctly and support their conclusions.

Reviewer #2 (Remarks to the Author):

Overall this is an interesting article submitted by Mensali and colleagues describing the specificity and antitumor activity of OSCAR T cells targeting ALPL-1 in osteosarcoma.

The work is particularly noteworthy and significant because patients suffering with osteosarcoma need novel therapies. For the field of CAR T cell therapy for osteosarcoma, there are relatively few surface antigens to target, and this work adds ALPL-1 to the list of potential CAR T cell targets in this patient population.

There are several strengths:

- The authors nicely support specificity of OSCAR T cells for ASPL-1, including both over-expression and knockout experiments.
- In vitro antitumor activity is nicely evaluated in multiple settings, including cell lines, 3D spheroids, and primary osteosarcoma samples.
- Multiple in vivo studies are performed to evaluate OSCAR T cell activity against osteosarcoma, including an intratibial model that also has lung disease.

There are also some weaknesses that may or may not be able to be addressed with additional experiments or modifications to the text.

Major Concerns:

- While OSCAR T cells clearly have antitumor activity in multiple animal models, the CAR T cells are given multiple times, at very high doses, and at early timepoints after tumor injection. Given these experimental designs, interpretation of the potency and potential utility of OSCAR T cells is questionable.
- Given the need to treat tumors in vivo with multiple high doses of CAR T cells, a paper of this caliber should explore alternative structural domains, costimulatory domains or other modifications to improve CAR T cell function. A final conclusion of the work is to support a clinical trial using OSCAR T cells, but the findings described here support further exploration to optimize CAR design before moving in such a direction.
- ALPL-1 protein expression is not well evaluated in this paper. Strong conclusions about utility of the CAR are made based on documentation of ALPL-1 protein expression, which should be further evaluated and documented in this manuscript. Furthermore, gene expression of ALPL-1 in OS compared to normal tissues in Fig 1E is not completely convincing, further establishing a need for protein evaluation in OS samples.
- A majority of in vitro experiments are done with only 1-2 donors, including an important cytokine analysis experiment (Fig 2F) done with only 1 donor. 3-4 donors or more should be used for all studies unless there is substantial reason to limit the number of donors.
- %CAR surface expression is not well established between OSCAR 1 and 3, limiting the interpretation of several experiments. Documentation of CAR expression in multiple donors would improve the ability to determine if differences in CAR T cell function are real or simply due to differences in CAR expression levels.

Minor Concerns/Suggestions/Comments:

- The introduction would benefit from succinctly describing what's known about ALPL/ALPL1 in OS disease and progression
- Line 97 states CARs are specific to cancer cells only, however they also recognized

osteoblasts/MSCs

- Statement of ALPL1 being the broadest and most specific should be removed or reworded. This statement is difficult to support, and no comparisons to other CAR targets are made in this paper.
- Please use primary research publication(s) to support ALPL-1 protein expression in osteosarcoma and that TP antibodies react with virtually all OS samples tested. Reference 8 is a review article. Given importance of ALPL-1 protein expression in osteosarcoma, cross reference primary manuscripts is essential.
- Pre-clinical work targeting GD2, EGFR, and B7-H3 using CAR T cells should be cited
- Many figure panels are too small and nearly illegible, example CAR schema in Fig 2A. Larger font and figure panels will help readers interpret results
- Line 154: Why were T cells listed as 'transfected' here?
- Ext Fig 4B,C – Total flux appears similar or above flux at day 3 for both treatment groups, however representative images show no evidence of tumor at the day 21 time point.
- Why was % TNF alpha positive T cells used to measure recognition of normal tissues? Cytokine expression similar to 2F would be a better measure and easier to interpret
- Recommend authors do not include statements on data not presented in the paper. Please remove statements stating data not included etc or otherwise add the data to main or extended data panels.
- The term functional avidity is not preferred by this reviewer (and not common in the field) because it could confuse readers with avidity or strength of multiple CAR/Antigen interactions, which were not specifically evaluated here. Simply state the effector function tested (Ca++ flux) or otherwise.
- Fig 3C shows flux measurements for multiple OSCAR T cell treated animals stops at approximately day 21, but 100% of mice in Fig 3D survive at this time point. Are these mice excluded from survival analysis? What is the explanation for their endpoint at day 21 - GVHD?
- Fig 3B and others - in vivo radiance/flux scale bars should include numbers used to define min & max signal
- Fig 4G shows many nearly all mice have tumor signal at days 24, 50 and 64 similar to signal on day 10, however 4F shows nearly all CAR treated mice have no tumor signal at these timepoints.
- Extended Figure 3C: 10:1 ratio against LM7 is very high. Evaluating lower E:T ratios would help readers determine potency of OSCAR T cells.
- Extended Figure 4C – individual radiance values are only shown to day approximately day 28, however survival up to 100 days. Please include full radiance data up to the study endpoint in this panel.
- Figures panels 1A, B: Figure legend colors do not match flow histograms.
- Line 202-203: Statement that metastases were controlled is too strong. Please rephrase to delayed tumor growth or exhibited antitumor activity etc. Fig 3G clearly shows some delayed growth followed by clear tumor progression in this model

Reviewer #3 (Remarks to the Author):

Fig 1E: Showing transcript levels is not sufficient to determine whether or not TP-1/TP-3 recognized ALPL1 is present on normal bone or lung. Were any IHC stains of normal tissue done with TP-1/TP-3? Your reference to prior testing of normal tissue is a review article- is there primary data to suggest that TP-1/TP-3 don't bind to normal tissue?

Fig 2B: This appears to be proliferation in the setting of CD3/CD28 bead activation, and not really relevant to the functionality of the CAR. Please show proliferation in the context of ALPL+ targets. Also, why do OSCAR-3 CAR T cells proliferate more than OSCAR-1 and mock- do OSCAR-3 cells tonically signal?

Fig 2D: Why not include this as another bar in Fig 2C?

Fig 2E: Please include the ALPL staining of these OS cell lines.

Fig 2F: What are the units of the 0-4 axis? What is the absolute amount of cytokine secreted by the cells? Will be more informative if you focused on showing just the relevant cytokine i.e. IL2, IL17, IFNg, and TNFa.

Fig 3D and 3H: Is there a statistically significant difference between the survival of OSCAR1 and

OSCAR3?

Fig 3D: Is there a statistically significant difference between the survival of OSCAR1 and OSCAR3?

Fig 3: Why are the outcomes different between OSCAR1 and OSCAR3 in the IP and IV models i.e. why is OSCAR1 curative in IP, but not in IV? It appears that OSCAR3 kills OHS more rapidly (Fig 2E) but this does not translate into a survival benefit in vivo- can you comment on this. Can you comment on the ALPL1 antigen density on the OHS vs LM7?

Fig 4: Do you have a survival curve for this xenograft?

Fig 5C: Why not include commercial ALPL antibody with TP1/TP3 antibodies on the same overlap figure- this allows for better comparison.

REVIEWER COMMENTS

Reviewer #1 (Remarks to the Author):

This is a well conducted and written study.

It is methodologically sound and all the experiments have been conducted correctly and support their conclusions.

We thank the Reviewer #1 for her/his positive comment on our work.

Reviewer #2 (Remarks to the Author):

Overall this is an interesting article submitted by Mensali and colleagues describing the specificity and antitumor activity of OSCAR T cells targeting ALPL-1 in osteosarcoma.

The work is particularly noteworthy and significant because patients suffering with osteosarcoma need novel therapies. For the field of CAR T cell therapy for osteosarcoma, there are relatively few surface antigens to target, and this work adds ALPL-1 to the list of potential CAR T cell targets in this patient population.

There are several strengths:

- The authors nicely support specificity of OSCAR T cells for ASPL-1, including both over-expression and knockout experiments.
- *In vitro* antitumor activity is nicely evaluated in multiple settings, including cell lines, 3D spheroids, and primary osteosarcoma samples.
- Multiple *in vivo* studies are performed to evaluate OSCAR T cell activity against osteosarcoma, including an intratibial model that also has lung disease.

There are also some weaknesses that may or may not be able to be addressed with additional experiments or modifications to the text.

We are grateful to Reviewer #2 for her/his encouraging comments and the appreciation of our efforts to demonstrate the interest of targeting ALPL-1 to treat osteosarcoma, we share her/his view about the lack of specific targets to treat this cancer.

We also agree that the first version of the manuscript contained some weaknesses and we appreciated her/his constructive remarks. We have provided specific responses to her/his comments and revised the manuscript accordingly. We hope that the enhanced version will satisfy her/him.

Major Concerns:

- While OSCAR T cells clearly have antitumor activity in multiple animal models, the CAR T cells are given multiple times, at very high doses, and at early timepoints after tumor injection. Given these

experimental designs, interpretation of the potency and potential utility of OSCAR T cells is questionable.

We agree with reviewer #2 that 10^7 CAR T cells is a high dose when compared to 10^6 CAR T cells or even less, 10^5 cells, which are now tested in several models, in particular in dose de-escalation studies or “stress-test” studies. However, injection of 10^7 CAR T cells has also been frequently used in experimental designs to test CAR anti-tumor activity/potency in solid cancers like in the following references from Nat Com⁴⁻⁶ or other prestigious journals^{7,8}. We also would like to point out the fact that we changed the doses with the model: we have used three doses in the OHS (Fig. 3a-d) and OSA (Extended Data Fig. 4) models and four doses in the LM7 model (Fig. 3e-h). However, two doses of CAR T cells were used in the intratibial model (Fig. 4) which is more related to what is usually presented. Importantly, low dose treatments are frequently used in well-established models such as leukemic models. We used these numbers of doses based on previous mouse models that we had tested and on the growth rate of the tumor, i.e. LM7 is slow growing we therefore wanted to avoid losing the effect of the Tc. In addition, since the OSCARs were never tested and osteosarcoma can be challenging to treat in a living system, our intention was to detect an effect and not to perform a “stress test” as reported by others in known models (Nalm-6 versus CD19CAR Tc)⁹. This was particularly relevant when we tested OSCARs in OSA which is a more aggressive and a fast-growing tumor model³ (Extended Data Fig. 4d) and has, to our knowledge, never been used to validate CAR T cell efficacy; Following this reasoning, we also injected a relatively high dose of tumor cells to ensure proper engraftment (typically $1-2 \times 10^6$ tumor cells), compared to other settings where mice received from 10^4 to 10^5 tumor cells; Another argument supporting our strategy to use high dose treatment was the fact that it was not toxic to the animal and the mock T cells did not affect the tumor growth or the health of the mice (xenoreactivity), suggesting that the detected efficacy was solely due to the CAR T cells. Finally, our strategy is in line with the application of the 3R principles which require a reduction of the animal experiments and the number of individuals per group (by testing lower inefficient doses).

Nevertheless, and in agreement with the reviewer comment, we are presently testing OSCAR-3 T cells co-expressing chemokine receptors for a follow up project and have addressed this dose issue using the OSA lung metastatic model (Joaquina et al. *in preparation*). Here the animals were injected with only two doses of OSCAR-3 T cells. The results show that two doses gave similar tumor control to what was observed with three; we include these data for the reviewers’ eyes only since they are part of another story in preparation. The experiment was performed as in the submitted paper and the data of mock Tc versus OSCAR-3 Tc (blue) are shown

Editorial Note: Figure created with illustrations from BioRender.com.

Thus these results confirm that the reviewer is correct and lower doses can be used, but also that our submitted *in vivo* experiments are still valid. We have clarified this point in the discussion (l. 291-295)

- Given the need to treat tumors *in vivo* with multiple high doses of CAR T cells, a paper of this caliber should explore alternative structural domains, costimulatory domains or other modifications to improve CAR T cell function. A final conclusion of the work is to support a clinical trial using OSCAR T cells, but the findings described here support further exploration to optimize CAR design before moving in such a direction.

We concur with reviewer#2 that exploiting alternative structural domains, costimulatory domains or other modifications is of paramount interest; however, as explained in the first answer and supported by the data added, lowering the dose of treatment did not alter OSCAR efficacy, thus the CD8h/TM-BBz might well be the final clinical design. We have previously tested the CD28h/TM-28z format and did not see any difference in *in vitro* assays (see below), we therefore focused on the first design because this is the one we use for most of our constructs¹⁰.

In the following experiments we compared the killing activity in BLI-based assays of T cell expressing OSCARs with different signaling tails. As shown, we did not detect any differences (n= 2 donors):

In addition, this is also supported by our new set of *in vitro* data requested (see next answer and Fig. 2) demonstrating that OSCARs are efficient and sensitive. Finally, we also feel that the comparison of alternative constructs is beyond the scope of this study. Here the focus is the pre-clinical validation of a novel target, ALPL-1, for CAR treatment in OS.

Concerning the validity of OSCARs for an eventual clinical proof-of-concept, we are now preparing the documentation where the OSCAR-3 will be tested in a transient setting (mRNA electroporated CAR T cells) rather than viral based expression. The main reason being safety since (i) this is a novel target and (ii) the patients will most likely be young adults or children. We have already tested *in vivo* OSCARs mRNA using a dose of 10^7 T cells to ensure a good pre-clinical validation of their efficacy. We observed tumor delayed growth in the animals:

We have kept the same *in vivo* settings for both transient (mRNA electroporation) and permanent (retroviral transduction) validation platform. These data will be integrated in the first-in-man report following the trial. This was already mentioned in the first version of the article (l. 361)

Although of importance, we think that the main factors that should be taken into account concerning the design of anti-osteosarcoma CARs are (i) combatting the very immunosuppressive tumor microenvironment (TME) and (ii) the humanization of the scFv. We are presently working on these points as mentioned (and shown) in the previous answer for the chemokine receptor combination and are comparing the original murine OSCAR with their humanized versions (*work in progress*). These are not meant to improve the construct *per se*, but rather to adapt it to its real-world environment.

We therefore agree with the reviewer that alternative designs are highly important and we are presently working on an adapted OSCAR. These improvements are not directly functional as we think that our 4-1BBz design is satisfactory, but they are mitigating the inhibitory effect of the immunosuppressive environment at the tumor site (TME) and on the immune side, potential anti-murine reactivity. We have added these points to the discussion (lines: 301-307)

- ALPL-1 protein expression is not well evaluated in this paper. Strong conclusions about utility of the CAR are made based on documentation of ALPL-1 protein expression, which should be further evaluated and documented in this manuscript. Furthermore, gene expression of ALPL-1 in OS compared to normal tissues in Fig 1E is not completely convincing, further establishing a need for protein evaluation in OS samples.

We agree that the protein description could have been better documented in the present paper. The reason for that was that the former reports on the TP antibodies description already provided a substantial amount of information^{1,2,11} about the antibody specificity. However, these stainings were performed before ALPL-1 was confirmed as the *bona fide* target. We have therefore decide to enhance the results documenting ALPL-1 expression at the protein level to support our conclusions. It is critical to indicate here that there are no commercial antibodies specific for ALPL-1 available and the one used is a general anti-ALPL antibody. It is therefore difficult to compare TP antibodies with an existing product. It is nevertheless important to show that ALPL antibody reacts against all TP target whereas TP antibodies appear more restrictive :

- 1) We have stained for ALPL protein on a larger panel of tumor cell lines with distinct origins (Fig. 5c), patient-derived xenografts (PDX) (Fig. 1f), and we have added more primary cells from healthy tissues (PBMCs, bone marrow, CD34+ cells from PBMCs, and HUVEC). Cells were stained side by side with TP-3 antibody or anti-ALPL antibody and analyzed by flow cytometry. As negative controls we have used secondary antibody staining for TP-3 staining and a matching isotype staining for ALPL staining. The results show that TP-3 binds specifically to OS derived tumor cells with high staining intensity in general, and no TP-3 binding was detected on other cell lines. Correspondingly, the anti-ALPL commercial antibody binds to OS cell lines following a similar pattern to TP-3 (Fig. 5c, Extended Data Figure 5), but also recognize some healthy tissues. Differences in intensity between TP-3 and ALPL staining can be explained by the fact that TP-3 is an indirect staining and anti-ALPL is a directly conjugated antibody. We have also observed that some cell lines or primary cells were positive for ALPL staining but not TP-3, which suggest the presence of other ALPL isoforms not detected by TP-3 in these tissues (l. 159-165).
- 2) We have performed immunohistological (IHC) staining of OS slides and healthy tissues and confirmed previous and recent work in humans and dogs^{1,11,12}. These data are important support of not only the presence of the antigen in cancer cells but also its restriction, described in the results (l. 165-168) and discussion (l. 284-287)

The results have been included in the manuscript in Fig. 1f-g and Fig. 5c, described in the Methods section (l.413-428), Results section (l.159-168), and mentioned in the Discussion section (l. 284-287). We have also improved the OSCAR target description in the Introduction section (l. 88-102)

- A majority of *in vitro* experiments are done with only 1-2 donors, including an important cytokine analysis experiment (Fig 2F) done with only 1 donor. 3-4 donors or more should be used for all studies unless there is substantial reason to limit the number of donors.

To address the reviewer's concerns and assess donor T cell variability we have generated CAR T cells from more healthy donors (4) and have repeated the most relevant *in vitro* validation experiments (Fig. 2). These include the cytokine profiling (Fig. 2e), we have also modified this figure upon request of Reviewer #3 and presented the cytokines she/he indicated.

We have checked for CAR expression (Fig. 2a shows CAR expression from 6 donors) and we have repeated T cell expansion on CD3/CD28 beads (Fig. 2b) shows proliferation from 6 donors), measured tumor cell lysis by BLI assay with different E:T ratios as previously mentioned (Figure 2d, 4 donors), and we have repeated the BLI cytotoxicity assay against ALPL knock in and knock out cells (Figure 2c, 3 donors) as well as measured cytokine release by bioplex assay (Figure 2e, 2 to 4 donors). The original Figure 2 has been substantially changed including these the new experiments and the text adapted accordingly (l.179-205). Importantly, this increase in the number of donors led us to the modification of a few observations: OSCAR Tc and mock Tc follow the same expansion pattern (Fig. 2b) and IL-2 release was more potent for OSCAR-3 Tc than OSCAR-1 Tc (Fig. 2e).

- %CAR surface expression is not well established between OSCAR 1 and 3, limiting the interpretation of several experiments. Documentation of CAR expression in multiple donors would improve the ability to determine if differences in CAR T cell function are real or simply due to differences in CAR expression levels.

Please, see the answer above. We have now addressed this by including CAR expression in the additional donors tested (Fig. 2a). CAR expression data measured by anti-CD34 staining did not reveal any statistical significance between OSCAR-1 and OSCAR-3. Unfortunately, the anti-Fab staining could not be used to measure differences in CAR expression between OSCAR-1 and OSCAR-3 because this antibody poorly stains OSCAR-1. We see two explanations: (i) the polyclonal anti-Fab antibodies were generated against a "variable" domain of the murine antibody, these can vary in their V chain type expression, which will impact the anti-Fab. OSCAR-1's V chain might be less represented, hence less detected. (ii) OSCAR-1 is less expressed or its protein less stable This will not be detected by using a 2A-surrogate marker like tCD34 (which only indicate whether the mRNA is transcribed). Nevertheless, the OSCAR-1 expression measured with anti-Fab never correlates with the functional response (activation, killing or *in vivo* efficacy) of OSCAR-1 T cells and we therefore have to rely on the surrogate marker (tCD34) which seems to more accurately correlate with functional response. Importantly, whether this is due to a poor detection or a lower expression doesn't appear as a main concern, indeed, the observed function of OSCAR-1 appears efficient despite the poor anti-Fab detection. Comments were added about that (l. 179-180 and l.301-307).

Minor Concerns/Suggestions/Comments:

- The introduction would benefit from succinctly describing what's known about ALPL/ALPL1 in OS disease and progression

We have added a succinct description of ALP and ALPL in the Introduction section (l.88-93 and l. 99-102)

- Line 97 states CARs are specific to cancer cells only, however they also recognized osteoblasts/MSCs

We apologize for the inaccuracy; we have now modified the text (l.110-111) in accordance with the reviewer's comment.

- Statement of ALPL1 being the broadest and most specific should be removed or reworded. This statement is difficult to support, and no comparisons to other CAR targets are made in this paper.

The statement has been reworded (l. 111-112)

- Please use primary research publication(s) to support ALPL-1 protein expression in osteosarcoma and that TP antibodies react with virtually all OS samples tested. Reference 8 is a review article. Given importance of ALPL-1 protein expression in osteosarcoma, cross reference primary manuscripts is essential.

We agree with the reviewer and we thank him/her for this comment. We apologize for the mistake that was made in the bibliography The ref.8 has now been replaced with the correct one which is an original research article¹ (l. 151.)

- Pre-clinical work targeting GD2, EGFR, and B7-H3 using CAR T cells should be cited

We agree with the reviewer. We previously cited the study of Ahmed N. on HER-2 CAR in sarcoma (l. 81-82) and have now included additional references for the other CARs mentioned¹³⁻¹⁶.

- Many figure panels are too small and nearly illegible, example CAR schema in Fig 2A. Larger font and figure panels will help readers interpret results

We have now improved the quality of the figures and have enlarged the font and the visibility. We hope that the reviewer finds them easier to read.

- Line 154: Why were T cells listed as 'transfected' here?

We thank the reviewer for pointing out this error, we meant "transduced" and the text has been corrected accordingly (l. 176).

- Ext Fig 4B,C – Total flux appears similar or above flux at day 3 for both treatment groups, however representative images show no evidence of tumor at the day 21 time point.

We agree with the reviewer that the representative images of the tumor on day 21 do not fit with the total flux measurements. This is due to the fact that the pictures on day 3 were taken with different instrument settings in order to help the reader to appreciate the tumor presence and also to avoid saturation at later time points. We apologize that this was not clear and we have now specified the difference in settings of the pictures in the figure by the inclusion of more specific scale bars.

- Why was % TNF alpha positive T cells used to measure recognition of normal tissues? Cytokine expression similar to 2F would be a better measure and easier to interpret

We partially agree with Reviewer #2 on this comment because the main purpose of the experiment in this specific context was to assess CAR safety by investigating CAR cross-reactivity towards cells from normal tissues. At this stage we did not want to characterize the full cytokine profile of our CAR T cell product; however, we think that cytokine characterization is important and, indeed we have

assessed the cytokine profile as showed in Fig. 2e. There are several ways to look for CAR T cell activation (effector cytokine release/expression, degranulation, tumor cell lysis, ect). TNFa, IFNg and the degranulation marker CD107a are the most broadly used as markers to indicate T cell activation, and, these are the markers that we typically assess by flow cytometry. Here TNFa was used to test recognition of healthy tissue as this is the most sensitive marker in our hands. TNFa is produced rapidly by both CD4 and CD8 T cells following CAR engagement and this fits with the short time of the *in vitro* assay (6 hours co-incubation). We used a flow cytometry-based assay because we wanted to look at cytokines at a single cell level and to separate the CD4 and CD8 T cell populations. In the first experiments we also included IFNg and CD107a and the expression profiles of these are similar to TNFa, so we therefore decided to use TNFa as the read-out.

- Recommend authors do not include statements on data not presented in the paper. Please remove statements stating data not included etc or otherwise add the data to main or extended data panels.

We agree with the reviewer#2 that this statement “data not shown” can be misleading, but our intention was to not include these data as they have no direct influence on our results. We have removed 2 of them, the first describing the non-recognition of ALPL isoform-3 and the last about the binding propriety. The second describing the inefficiency of TP antibodies to work in Western blot is important for the reader, but the plots have no interest since they are empty, so we respectfully ask to keep only this reference (l. 131).

- The term functional avidity is not preferred by this reviewer (and not common in the field) because it could confuse readers with avidity or strength of multiple CAR/Antigen interactions, which were not specifically evaluated here. Simply state the effector function tested (Ca++ flux) or otherwise.

The term functional avidity was changed and replaced with “slightly more potent in triggering the release of some of the cytokines”(l. 195-196).

- Fig 3C shows flux measurements for multiple OSCAR T cell treated animals stops at approximately day 21, but 100% of mice in Fig 3D survive at this time point. Are these mice excluded from survival analysis? What is the explanation for their endpoint at day 21 - GVHD?

We thank the reviewer for her/his comment and apologize for this mistake, we have now included all the pictures after day 21 (Figure 3b).

- Fig 3B and others - *in vivo* radiance/flux scale bars should include numbers used to define min & max signal

This has been done in all figures with *in vivo* studies. We would like to thank the reviewer for this comment, because while we were correcting the scales, our collaborator in Barcelona realized that their analyse had not been performed at the same scale, consequently we also replace the pictures; this does not change the results/conclusions, but rather improves the appreciation of the OSCAR effect (Figure 4b).

- Fig 4G shows many nearly all mice have tumor signal at days 24, 50 and 64 similar to signal on day 10, however 4F shows nearly all CAR treated mice have no tumor signal at these timepoints.

The reviewer is correct that different settings were used for the day 10 IVIS image in **Figure 3g** (there was no Figure 4G) to visualize tumor engraftment before T cell injection. As scale bars with numbers have now been included this has now been indicated and we hope that this is clearer.

- Extended Figure 3C: 10:1 ratio against LM7 is very high. Evaluating lower E:T ratios would help readers determine potency of OSCAR T cells.

We agree with the reviewer and we have performed a new set of cytotoxicity assays with several donors where we have tested different E:T ratios and show that the T cells can still kill efficiently at lower E:T ratios. (**Figure 2d**).

- Extended Figure 4C – individual radiance values are only shown to day approximately day 28, however survival up to 100 days. Please include full radiance data up to the study endpoint in this panel.

The wrong survival curve was inserted with this figure, the correct graph showing shorter survival was added.

- Figures panels 1A, B: Figure legend colors do not match flow histograms.

This has been corrected in the current version of the manuscript.

- Line 202-203: Statement that metastases were controlled is too strong. Please rephrase to delayed tumor growth or exhibited antitumor activity etc. Fig 3G clearly shows some delayed growth followed by clear tumor progression in this model

We agree with the reviewer and we have modified the sentence (**l. 231-232**)

Reviewer #3 (Remarks to the Author):

We thank the reviewer #3 for her/his careful reading of our manuscript and feedback.

Fig 1E: Showing transcript levels is not sufficient to determine whether or not TP-1/TP-3 recognized ALPL1 is present on normal bone or lung. Were any IHC stains of normal tissue done with TP-1/TP-3? Your reference to prior testing of normal tissue is a review article- is there primary data to suggest that TP-1/TP-3 don't bind to normal tissue?

We agree with the reviewer #3 and we thank him/her for this comment. We apologize because the review reference was mistakenly added instead of the original research article The ref.8 has now been replaced with the correct one which is an original research article¹ (**l. 151.**)

We agree that showing transcript levels of the target is not sufficient to exclude recognition of ALPL-1 on normal tissues by TP antibodies. To support prior data on TP binding of normal tissues we have now provided a new set of data where we stained a broad spectrum of tumor cell lines with different origin, PDX from 7 patients (**Fig. 1f**), and cells from normal tissues with TP-3 and, in parallel, with commercial anti-ALPL antibody (since no commercial anti-ALPL-1 is available, **Fig. 5c**). We also performed some IHC staining on tumor and healthy tissues suspected to express ALPL (**Fig. 1g**). Due to the difficulty of finding cells representative of distinct healthy tissues we have used, as an

alternative, tumor cell lines of diverse origin to represent the different tissues, even if this might look as an approximation, we remind the reviewer that primary healthy tissues were previously tested and included in the first version (Fig. 5a-b). They included lung, liver, kidney (pulmonary alveolar epithelial cells, hepatocytes, renal epithelial cells) and we have now added PBMC, BM, HUVEC.

The results confirm our previous observations and reinforce the safety profile of the OSCAR constructs.

Fig 2B: This appears to be proliferation in the setting of CD3/CD28 bead activation, and not really relevant to the functionality of the CAR. Please show proliferation in the context of ALPL+ targets. Also, why do OSCAR-3 CAR T cells proliferate more than OSCAR-1 and mock- do OSCAR-3 cells tonically signal?

The proliferation capacity of CAR T cells in expansion conditions mimicking the setup of the clinical manufacturing is a commonly used method to demonstrate that the CAR presence does not alter the T cell division. It is very frequently shown at the beginning of the publication as a proof of non-toxicity of the CAR. Thus, the intention here is to show that OSCAR T cells are viable and could potentially be produced in large scale.

Concerning the reviewer #3 comment about the increased proliferation capacity of OSCAR-3 Tc due to receptor tonicity, we decided to test this by increasing the number of donors and the new Figure 2b shows that OSCAR-3 Tc performed similarly as OSCAR-1 and mock Tc. Thus, the difference detected in the first version was probably due to the fact that only 2 donors were used. The text was modified (l. 181-182) and we thank the reviewer for pointing this out.

As requested by Reviewer #3, we performed a re-challenge assay where CAR T cells from 4 donors were co-cultured in the presence of irradiated ALPL-1+ cells (OHS) used as feeder cells. In this *in vitro* assay we investigated long-term cytotoxicity (Fig. 2f) and proliferative capacity of CAR T cells over 28 days, at intervals of 7 days, upon addition of fresh irradiated ALPL-1+ tumor targets (rechallenge) for a total of 3 cycles: (data not included in the article)

Our results indicate that OSCAR T cells maintained robust short-term effector function as they mediated effective elimination of OS cells until day 21 which corresponded to 3 rechallenges (Fig. 2f). Repetitively stimulated CAR T cells also maintained their proliferative capacity up to the 2nd re-

challenge but then proliferated less (see figure below) and we therefore had very few cells to test on d28 after the 3rd rechallenge. CAR expression remained stable during the re-challenge as shown here (data not included in the article).

Together these data support the OSCAR-dependent proliferative capacities on ALPL positive target. We think that showing the maintained killing capacity of OSCAR T cells is more informative than the expansion on feeder cells.

Fig 2D: Why not include this as another bar in Fig 2C?

As suggested by Reviewer #3, we have included the data of Fig. 2d in the bar graph of the Fig. 2c, we agree that this improves the clarity of the message. Fig. 2d has been replaced by the killing assay testing different E:T ratios.

Fig 2E: Please include the ALPL staining of these OS cell lines.

This has been done and is shown in the new Fig. 5c that includes the staining of different cell lines. We here include the flow cytometry plots for the reviewer

Fig 2F: What are the units of the 0-4 axis? What is the absolute amount of cytokine secreted by the cells? Will be more informative if you focused on showing just the relevant cytokine i.e. IL2, IL17, IFN γ , and TNF α .

We agree with the reviewer that the heatmap, although frequently used by others¹⁷, might be confusing. We have therefore replaced it with a new graph (Fig. 2e) to show only relevant effector cytokines including IL-17 as suggested. The numbers shown are the observed concentrations of

cytokine (pg/mL). We have also increase the number of donors. We hope that these modifications will satisfy the Reviewer.

Fig 3D and 3H: Is there a statistically significant difference between the survival of OSCAR1 and OSCAR3?

We thank the reviewer for pointing out these missing data. The statistical points were omitted and we have corrected this. Fig. 3d: no statistical significance was measured between the survival of mice treated with OSCAR-1 and OSCAR-3 Tc. Fig. 3h: statistical significance was reached between OSCAR-1 and OSCAR-3

Fig 3D: Is there a statistically significant difference between the survival of OSCAR1 and OSCAR3?

We answered this question above.

Fig 3: Why are the outcomes different between OSCAR1 and OSCAR3 in the IP and IV models i.e. why is OSCAR1 curative in IP, but not in IV? It appears that OSCAR3 kills OHS more rapidly (Fig 2E) but this does not translate into a survival benefit *in vivo*- can you comment on this. Can you comment on the ALPL1 antigen density on the OHS vs LM7?

We have noticed that OSCAR-3 expressing T cells seem to be more potent in *in vitro* experiments. Our data suggest that OSCAR-3 T cells kill faster and probably produce more cytokines (e.g. IL-2), however, in a long term assay, as for the *in vivo*, OSCAR-1 T cells catch up and perform as well as OSCAR-3 T cells. We have also performed an antigen-density detection on different cell lines but could not do it for LM7 due to the presence of GFP in our strain; the kit only allows FITC detection. We chose to however run OSA and OHS together with non-OS cell lines. The resulting demonstrates a presence of a high amount of ALPL-1 on the surface of these cells, but the density does not correlate with the killing of these cells with OSCARs, this has been added in the text (Extended Data Fig. 1d, l. 198-200)

Fig 4: Do you have a survival curve for this xenograft?

We do not have a survival curve for the xenograft experiment because the humane endpoint was reached for other reasons and the mice euthanized before the animals died of the cancer, this was indicated in the Methods section (l. 640) and has been added in the figure (l. 923).

Fig 5C: Why not include commercial ALPL antibody with TP1/TP3 antibodies on the same overlap figure- this allows for better comparison.

We agree that this would allow a proper comparison. We did not use overlapping plots of TP1/TP3 and ALPL staining because ALPL is a directly conjugated antibody while the TP1/TP3 antibodies are not conjugated and require a secondary antibody. Hence the staining protocols were different. Additionally, different fluorochromes were used to detect ALPL (PE) and TPs (APC) and, usually secondary staining gives a brighter fluorescence intensity signal. This figure has been updated and confirmed that TP-3 restriction to OS cell line, whereas anti-ALPL is more cross-reactive.

References:

1. Bruland, O., Fodstad, O., Funderud, S. & Pihl, A. New monoclonal antibodies specific for human sarcomas. *Int. J. Cancer* **38**, 27–31 (1986).

2. Bruland, O. S., Fodstad, O., Stenwig, A. E. & Pihl, A. Expression and characteristics of a novel human osteosarcoma-associated cell surface antigen. *Cancer Res.* **48**, 5302–5309 (1988).
3. Lauvrak, S. U. *et al.* Functional characterisation of osteosarcoma cell lines and identification of mRNAs and miRNAs associated with aggressive cancer phenotypes. *Br. J. Cancer* **109**, 2228–2236 (2013).
4. Li, H. *et al.* Targeting brain lesions of non-small cell lung cancer by enhancing CCL2-mediated CAR-T cell migration. *Nat Commun* **13**, 2154 (2022).
5. Moghimi, B. *et al.* Preclinical assessment of the efficacy and specificity of GD2-B7H3 SynNotch CAR-T in metastatic neuroblastoma. *Nature Communications* **12**, 511 (2021).
6. Tseng, H. *et al.* Efficacy of anti-CD147 chimeric antigen receptors targeting hepatocellular carcinoma. *Nat Commun* **11**, 4810 (2020).
7. Ma, L. *et al.* Enhanced CAR-T cell activity against solid tumors by vaccine boosting through the chimeric receptor. *Science* **365**, 162–168 (2019).
8. Reinhard, K. *et al.* An RNA vaccine drives expansion and efficacy of claudin-CAR-T cells against solid tumors. *Science* **367**, 446–453 (2020).
9. Hamieh, M. *et al.* CAR T cell trogocytosis and cooperative killing regulate tumor antigen escape. *Nature* **568**, 112–116 (2019).
10. Köksal, H. *et al.* Chimeric antigen receptor preparation from hybridoma to T-cell expression. *Antib Ther* **2**, 56–63 (2019).
11. Haines, D. M. & Bruland, O. S. Immunohistochemical detection of osteosarcoma-associated antigen in canine osteosarcoma. *Anticancer Res* **9**, 903–907 (1989).
12. Kerboeuf, M. *et al.* Early immunohistochemical detection of pulmonary micrometastases in dogs with osteosarcoma. *Acta Vet Scand* **63**, 41 (2021).
13. Ahmed, N. *et al.* Human Epidermal Growth Factor Receptor 2 (HER2) -Specific Chimeric Antigen Receptor-Modified T Cells for the Immunotherapy of HER2-Positive Sarcoma. *J. Clin. Oncol.* **33**, 1688–1696 (2015).
14. Heczey, A. *et al.* CAR T Cells Administered in Combination with Lymphodepletion and PD-1 Inhibition to Patients with Neuroblastoma. *Mol Ther* **25**, 2214–2224 (2017).
15. Charan, M. *et al.* GD2-directed CAR-T cells in combination with HGF-targeted neutralizing antibody (AMG102) prevent primary tumor growth and metastasis in Ewing sarcoma. *Int J Cancer* **146**, 3184–3195 (2020).
16. Ahmed, N. *et al.* HER2-Specific Chimeric Antigen Receptor-Modified Virus-Specific T Cells for Progressive Glioblastoma: A Phase 1 Dose-Escalation Trial. *JAMA Oncol* **3**, 1094–1101 (2017).
17. Nix, M. A. *et al.* Surface Proteomics Reveals CD72 as a Target for In Vitro-Evolved Nanobody-Based CAR-T Cells in KMT2A/MLL1-Rearranged B-ALL. *Cancer Discov* **11**, 2032–2049 (2021).

REVIEWER COMMENTS

Reviewer #1 (Remarks to the Author):

Thanks to the Author for their reply. No further questions.

Reviewer #2 (Remarks to the Author):

The major concerns have been addressed in this revised manuscript and I commend the authors for this novel work.

Reviewer #3 (Remarks to the Author):

Excellent job by the authors- most of my concerns have been addressed other than for what is mentioned below. I believe the additional questions will be asked by many of us involved in getting CAR T cells to the clinic, and are necessary to get a better understanding on how to optimize CAR T cell development.

Fig 1E: For completeness, the authors should consider also showing TP-1 antibody data in Fig 1f and g.

Fig 2B: The capacity of CAR T cells to expand correlates with their in vivo efficacy. The data demonstrates that the OSCAR T cells do secrete cytokines and are activated with addition of OSA cells (Ca⁺⁺ mobilization), but the data demonstrating expansion is not compelling- OSCAR-3 has a 50% expansion increase over mock T cells, while OSCAR-1 has no difference. Importantly, mock T cells were used here, and not an off-target CAR T cell e.g. anti-CD19 or anti-CD33, as the latter would control for the presence of a costimulatory domain i.e. expansion seen could be simply due to tonic signaling. An alternative to show expansion is the use of target cells with ALPL1 knocked out (you do have these cells: OHS-ALPL KO), or the use of 293-HEK cells ectopically expressing ALPL1. Could the limited expansion have downstream consequences such as limiting the in vivo efficacy of these cells- 10e7 x3-4 treatments are needed to control xenograft tumors, and that is a lot of cells.

Fig 2E: Why not also include TP1 staining in Fig 5c?

Fig 2F: Please show excluded IFNg as although high, would be informative and a strong indication of the CAR T functionality.

Fig 3: Do OSCAR-3 get exhausted faster than OSCAR-1? This could explain differences observed short-term vs long-term, as well as in vivo efficacy? An in vitro assay, or exhaustion profiling on CAR T cells harvested from OHS (day 20) or LM7 (day 60) could elucidate the reason for differing efficacy.

Fig 5C: There are commercial conjugation kits available in most fluorochromes.

REVIEWER COMMENTS

Reviewer #1 (Remarks to the Author):

Thanks to the Author for their reply. No further questions.

Reviewer #2 (Remarks to the Author):

The major concerns have been addressed in this revised manuscript and I commend the authors for this novel work.

We thank the Reviewers #1 and #2 for their previous comments, which have led to an improved manuscript.

Reviewer #3 (Remarks to the Author):

Excellent job by the authors- most of my concerns have been addressed other than for what is mentioned below. I believe the additional questions will be asked by many of us involved in getting CAR T cells to the clinic, and are necessary to get a better understanding on how to optimize CAR T cell development.

We are grateful to Reviewer #3 for the appreciation of our efforts to address her/his concerns and we thank her/him for the further comments which will definitely help us to prepare the clinical translation of OSCAR.

We have provided specific responses to her/his comments and revised the text accordingly, hoping that this version will satisfy her/him.

Fig 1E: For completeness, the authors should consider also showing TP-1 antibody data in Fig 1f and g.

We understand the concerns of the reviewer about the completeness of the study. We did not include TP-1 staining in Figs. 1f, g and 5c for the following reasons:

- Firstly, in addition to the financial and material concerns and as indicated at the end of the discussion section, OSCAR-3 was selected for a first-in-human study (line: 373-376) and we therefore organized this expensive and time consuming analysis with the TP-3 antibody only. Note that the staining presented will also be used as supporting material for the clinical protocol regulatory approval.
- Secondly, considerable information on the two TP antibodies for tissue-staining was already provided in earlier studies¹⁻³ and was further supported in the present paper by new experiments (Figs. 1a, b and d and Extended Data Fig. 5). We have herein confirmed that TP antibodies recognize the same target, ALPL-1, (Extended Data

Figure 1a) and have a similar binding pattern for both OS cell lines (Fig 1a, b, d) and healthy cells (Extended Data Figure 5), with slightly higher binding intensity for TP3 than TP1. Thus staining with TP-3 might reveal low level ALPL-1 expression not detected by TP-1. In addition, we have shown a comparable activation profile of T cells expressing OSCAR constructs towards different targets (Fig. 2). At this stage of the OSCAR development, and because of TP-3 and TP-1 have demonstrated a similar recognition pattern², we believe that it justifies the use of TP-3 only, again, considering the limited availability of some of the primary healthy cells, tissues and PDX. By doing this we could broaden the investigation of TP-3 binding profile to a larger panel of tumor and healthy tissues, which we believe is important in terms of safety before moving to clinical testing.

- Finally, although we agree on the principle of testing TP-1 antibody, provided that we will further develop it for clinical use, we believe that it is legitimate to question the validity of comparing the binding of an antibody to the one of a CAR which is in fact designed from a single chain Fragment variable (scFv). In addition, we are presently working on humanized version of the OSCARs, thus, will TP-1 antibody still reflect the binding of these molecules? If the screen is meant to detect ALPL-1 expression in healthy tissues, then as argued before, TP-3 should be even more sensitive, thus the answer is provided herein. Furthermore, TP-1 was previously injected in humans without demonstrating toxicity⁴. If the objective is to detect cross-reactivity, then it might be wiser to invest in the production of (humanized) TP-1 scFv and screen tissues rather than testing TP-1, which might be a poor predictor of the CAR cross-reactivity. This is an important point that is very rarely discussed. We included preliminary staining data using scFv of TP-1 and TP-3 for the Reviewers only which supports the lower binding capacities of TP-1 also as in the scFv format (Jurkat are ALPL-1 negative and OHS is the test):

We have added clarifications concerning our choice to restrict the staining to TP-3 antibody (lines: 160-162), hoping that Reviewer #3 will understand and agree with our justifications.

Fig 2B: The capacity of CAR T cells to expand correlates with their in vivo efficacy. The data demonstrates that the OSCAR T cells do secrete cytokines and are activated with addition of OSA cells (Ca⁺⁺ mobilization), but the data demonstrating expansion is not compelling-

OSCAR-3 has a 50% expansion increase over mock T cells, while OSCAR-1 has no difference. Importantly, mock T cells were used here, and not an off-target CAR T cell e.g. anti-CD19 or anti-CD33, as the latter would control for the presence of a costimulatory domain i.e. expansion seen could be simply due to tonic signaling. An alternative to show expansion is the use of target cells with ALPL1 knocked out (you do have these cells: OHS-ALPL KO), or the use of 293-HEK cells ectopically expressing ALPL1. Could the limited expansion have downstream consequences such as limiting the *in vivo* efficacy of these cells- $10^7 \times 3-4$ treatments are needed to control xenograft tumors, and that is a lot of cells.

We concur with Reviewer #3 that the increased expansion of OSCAR T cells compared to mock T cells could be in part sustained by receptor tonicity. To address these concerns and confirm ALPL-1 dependent proliferative capacity of OSCARs, we have repeated the *in vitro* expansion assay by including an irrelevant CAR construct, anti-CD19 CAR (clone FMC63 that we have previously used^{5,6}), as suggested by Reviewer #3, and performed a re-challenge assay where CAR T cells (OSCAR-1/-3 and CD19 CAR T cells) from four donors were co-cultured with irradiated 293-HEK feeder cells ectopically expressing (or not) ALPL-1. In this new *in vitro* assay, we monitored the proliferative capacity of OSCAR T cells over a total of three cycles, at intervals of five days, with freshly irradiated target cells (three re-challenges) by using the CellTrace™ Violet (CTV) fluorescent dye-based proliferation assay. Briefly, T cells were labeled with CTV at the start of each re-challenge before seeding them for co-culture in the presence of targets and, T cell proliferation was tracked by analyzing the CTV fluorescence at the end of the re-challenge (reduction of CTV fluorescence indicates proliferation). Our results confirmed OSCAR-dependent expansion on ALPL-1 positive target and the ability of OSCAR T cells to maintain the proliferative capacity upon repeated antigen exposure (Fig. 2g and Extended Fig. 2a, b and lines: 206-212 and line:309) indicating efficacy *in vivo*. In agreement with the reviewer comment on the high number of cells and doses we have previously shown (answer to Reviewer #2, page 2 of the former rebuttal) that we could observe similar *in vivo* efficacy of OSCAR-3 after reducing the number of injected T cells and reducing it to two doses.

As previously mentioned, T cells harvested from the first re-challenge were also co-cultured with irradiated wild-type HEK cells lacking the ALPL-1 antigen and T cell proliferation was measured by CTV:

Our results indicated that CAR T cells did not divide in the absence of the specific target. Instead, OSCAR T cells showed significant increased proliferation compared to the irrelevant CAR T cells when co-cultured in the presence of ALPL-1 positive targets suggesting that OSCAR proliferation was mainly dependent on the presence of ALPL-1. We cannot exclude that tonic signaling contributes at least in part to cell expansion, but it appears that the main increase is due to the CAR target recognition; of importance, antigen-dependent proliferation was detected after up to three re-challenges (Fig. 2g). Because of limited proliferation in the absence of ALPL, not enough T cells could be harvested to set the third re-challenge of co-culture with HEK-WT. Together these data support the ability of OSCAR T cells to maintain a substantial antigen-driven proliferation capacity which is necessary for enhancing *in vivo* efficacy.

Fig 2E: Why not also include TP1 staining in Fig 5c?

Please see our first comment.

Fig 2F: Please show excluded IFNg as although high, would be informative and a strong indication of the CAR T functionality.

It is not clear to us which Figure Reviewer#3 is referring to as Fig. 2f shows the rechallenge experiment. In Fig. 2e of the last version of our article, we provided data on the most relevant cytokines, including IFNg (upper right quadrant), in addition to TNFa, IL2 and IL17, as requested in the first revision. These data clearly displayed the concentration of cytokines released by OSCAR T cells in the presence of ALPL-1+ targets. We agree with the reviewer that IFNg is an important effector cytokine and indicative of CAR T cell functionality.

In support of this statement, we herein show (for the reviewer) the parallel results from the TNFa analysis in Fig. 5b, which also support IFNg release against different ALPL-1 positive samples. We had previously decided to keep only one cytokine because we thought that these data were overlapping. Overall, our *in vitro* data demonstrate that OSCAR T cells produce important effector cytokines, including IFNg and TNFa, upon CAR engagement with ALPL. We have included a reference to these data in the text (line: 273).

Fig 3: Do OSCAR-3 get exhausted faster than OSCAR-1? This could explain differences observed short-term vs long-term, as well as in vivo efficacy? An in vitro assay, or exhaustion profiling on CAR T cells harvested from OHS (day 20) or LM7 (day 60) could elucidate the reason for differing efficacy.

This is a valid statement and we thank the Reviewer to point it out. We have herein reported and discussed that OSCAR-3 T cells get activated faster and seem to be more potent *in vitro* and in short assays (Lines: 304-317). This is the reason why OSCAR-3 was selected for further clinical development in a transient format: since the molecule will disappear after ca 3 days, we need an effective product (discussed in line: 309, 311-315 and 374-380).

When we analyzed the first *in vivo* experiment, we were surprised by the potency of OSCAR-1 since it was quite “soft” *in vitro*. We agree with Reviewer #3 that due to this higher baseline activation, OSCAR-3 T cells might get exhausted faster than OSCAR-1 T cells. Trying to address this question we have performed a new *in vitro* experiment where we assessed the exhaustion profile of OSCAR-3 and OSCAR-1 T cells (from four donors) following three rounds of rechallenge in the presence of ALPL-1 positive feeder cells. After the third rechallenge OSCAR T cells were stained with anti-TIGIT, -PD-1, -TIM3 and -LAG3 antibodies. Overall, OSCAR T cells did not show a particularly exhausted phenotype as indicated by negative expression of PD-1 and LAG3 inhibitory receptors on both CD4 and CD8 T cells. CD8 T cells showed higher levels of TIM3 and TIGIT compared to irrelevant CAR (CD19CAR) which suggests that these markers were triggered in an antigen-specific manner:

As shown here, no significant differences were observed between OSCAR-3 and OSCAR-1 T cells. The re-challenge data (Fig. 2g) indicated that OSCAR-1 might be more resistant to multiple stimulations, but the reason for this could not be found in the exhaustion markers.

In addition, we harvested the lungs and blood from mice engrafted with OSA cells where we analyzed the presence of OSCAR T cells in single cell suspensions by flow cytometry and could not detect any difference between the constructs. Here we determine the presence of T cells (CD45+ CD3+) and analyzed the expression of PD-1. Thus, the *in vivo* analysis indicates the presence/" infiltration" of T cells in samples from both OSCAR-1 and OSCAR-3 treated mice but no difference in PD-1 expression, suggesting that OSCAR-3 T cells were not more exhausted than OSCAR-1. We are aware that further analysis needs to be performed before concluding, but at this stage we believe that the main driver of the discrepancy between *in vitro* and *in vivo* data is related to the lower functional avidity of OSCAR-1.

We feel that these data (immune checkpoint expression and homing *in vivo*) do not bring new information about the difference between the two constructs. We thus think that for sake of clarity and fluidity, they should be kept out of the manuscript.

Fig 5C: There are commercial conjugation kits available in most fluorochromes.

We thank the Reviewer for this information, as previously mentioned we will rather test (humanized) scFv of the TP constructs and we will label these.

References:

1. Bruland, O. S., Fodstad, O., Stenwig, A. E. & Pihl, A. Expression and characteristics of a novel human osteosarcoma-associated cell surface antigen. *Cancer Res.* **48**, 5302–5309 (1988).
2. Bruland, O., Fodstad, O., Funderud, S. & Pihl, A. New monoclonal antibodies specific for human sarcomas. *Int. J. Cancer* **38**, 27–31 (1986).
3. Kerboeuf, M. *et al.* Early immunohistochemical detection of pulmonary micrometastases in dogs with osteosarcoma. *Acta Vet Scand* **63**, 41 (2021).
4. Bruland, O. S. *et al.* Immunoscintigraphy of bone sarcomas--results in 5 patients. *Eur J Cancer* **30A**, 1484–1489 (1994).
5. Köksal, H. *et al.* Combinatorial CAR design improves target restriction. *J Biol Chem* **296**, 100116 (2021).
6. Köksal, H. *et al.* Preclinical development of CD37CAR T-cell therapy for treatment of B-cell lymphoma. *Blood Adv* **3**, 1230–1243 (2019).

REVIEWERS' COMMENTS

Reviewer #2 (Remarks to the Author):

No additional comments or questions.